# Revisiting Diffusion Models: From Generative Pre-training to One-Step Generation

**Bowen Zheng** [1]  **Tianming Yang** [1]

## Abstract

Diffusion distillation is a widely used technique to reduce the sampling cost of diffusion models, yet it often requires extensive training, and the student performance tends to be degraded. Recent studies show that incorporating a GAN objective may alleviate these issues, yet the underlying mechanism remains unclear. In this work, we first identify a key limitation of distillation: mismatched step sizes and parameter numbers between the teacher and the student model lead them to converge to different local minima, rendering direct imitation suboptimal. We further demonstrate that a standalone GAN objective, without relying a distillation loss, overcomes this limitation and is sufficient to convert diffusion models into efficient one-step generators. Based on this finding, we propose that diffusion training may be viewed as a form of generative pre-training, equipping models with capabilities that can be unlocked through lightweight GAN fine-tuning. Supporting this view, we create a one-step generation model by fine-tuning a pre-trained model with 85% of parameters frozen, achieving strong performance with only 0.2M images and near-SOTA results with 5M images. We further present a frequency-domain analysis that may explain the one-step generative capability gained in diffusion training. Overall, our work provides a new perspective for diffusion training, highlighting its role as a powerful generative pre-training process, which can be the basis for building efficient one-step generation models.

## 1. Introduction

Diffusion Models (DMs) (Song & Ermon, 2020a;b; Ho et al., 2020) have emerged as powerful tools for generative tasks, such as image and video synthesis, surpassing traditional methods such as Generative Adversarial Networks (GANs) (Goodfellow et al., 2014) in terms of sample quality. However, the generation process in DMs is slow and requires a series of iterative steps that impose significant computational overhead.

To speed up the generation process, several distillation techniques (Salimans & Ho, 2022; Gu et al., 2023; Song et al., 2023; Yin et al., 2023; Zhou et al., 2024a;c; Hsiao et al., 2024) have been proposed to reduce the multi-step diffusion process into a few- or one-step model. While promising, these methods necessitate large computational resources for the distillation and often fall short of matching the performance of the original multi-step diffusion models. However, recent advances (Sauer et al., 2023; Xu et al., 2023; Kang et al., 2024; Sauer et al., 2024; Li et al., 2024a; Kim et al., 2024a) have shown that incorporating a GAN objective in the distillation process can improve efficiency, demonstrating the potential of GAN-based 'distillation' approaches.

In this work, we first identify a fundamental limitation of existing distillation methods: the local minima of the teacher and student models may differ significantly, hindering effective knowledge transfer. In contrast, the GAN objective naturally circumvents this issue. Building on this insight, we propose D2O (Diffusion to One-Step), a novel approach that relies solely on an exclusive GAN objective, eliminating the need for instance-level distillation losses. D2O achieves competitive performance with drastically reduced data requirements, while conventional distillation methods demand vast training images when trained without a GAN objective. This significant reduction suggests that the D2O model is likely not learning from scratch, as that typically requires tens or millions of images.

Based on these results, we propose that the diffusion objective can be viewed as a generative pre-training process, wherein the model learns generalized generative capabilities that can be rapidly adapted to downstream tasks. To verify this idea, we create the D2O-F model, with most parameters

[1] Institute of Neuroscience, Key Laboratory of Brain Cognition and Brain-inspired Intelligence Technology, Center for Excellence in Brain Science and Intelligence Technology, Chinese Academy of Sciences, Shanghai, China. Correspondence to: Tianming Yang <tyang@ion.ac.cn>.

*Proceedings of the $42^{nd}$ International Conference on Machine Learning*, Vancouver, Canada. PMLR 267, 2025. Copyright 2025 by the author(s).

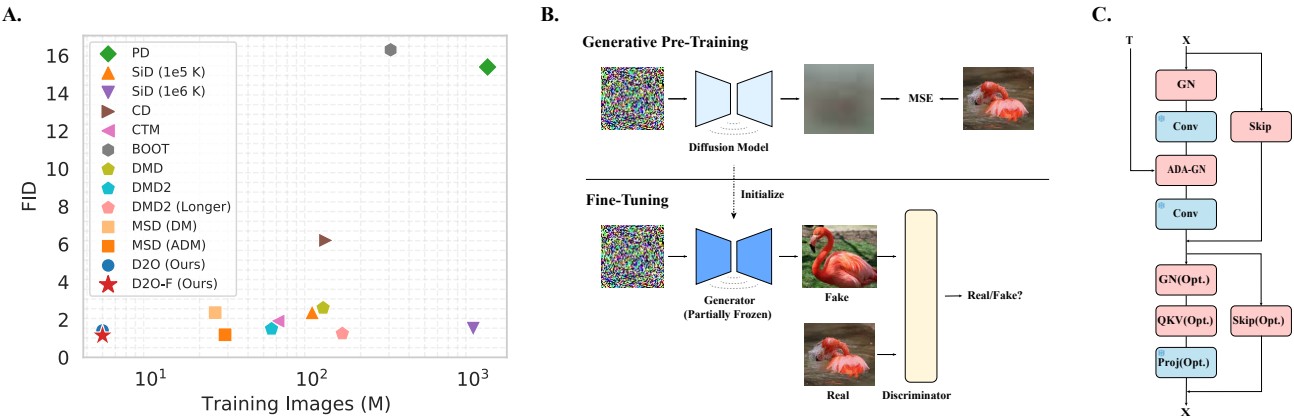

*Figure 1.* **A**. Comparison between our methods (D2O and D2O-F) and other methods on ImageNet 64x64. Our models show competitive results with a much smaller training set than the competing models. **B.** D2O-F model. We initialize the generator with a pre-trained diffusion model and freeze most convolutional layers during the fine-tuning. A simple GAN objective is adopted. **C.** A detailed illustration of the freezing method used in D2O-F blocks. Most convolutional layers (blue) are frozen. Only the normalization layer and the skip connection (red) are fine-tuned.

frozen during the fine-tuning. The D2O-F model shows high efficiency similar to D2O and exhibits even better performance. The success of the freezing method supports our theory that diffusion models possess universal generative capabilities and can be efficiently converted into an efficient one-step generator with a GAN objective.

## 2. Potential Limitation of Distillation Methods

### 2.1. Diffusion Models

There are several ways to interpret diffusion models, such as the score-matching approach (Song & Ermon, 2020a) and the probabilistic perspective (Ho et al., 2020). For the training process, given a clean image $x_0$ and random noise $\epsilon$, the training objective typically involves combining these with varying proportions and asking the model to predict the clean image. This task is equivalent to predicting the noise or the differences between image and noise.

In the EDM framework (Karras et al., 2022), a clean image $x_0$ is perturbed by adding Gaussian noise $x_{t_i} \sim \mathcal{N}(x_0, t_i^2 I)$, where $t_i$ is the standard deviation determined by a time-step scheduling function $T(i, N)$. Here, $N$ denotes the total number of discrete time steps, and $i$ refers to the current step index. A score model $\mathbf{g}_\phi(x_{t_i}, t_i)$ is then tasked with predicting the clean image $x_0$ using this perturbed image.

With this pre-trained score model, an ODE solver $\mathbf{S}_\phi$ can be defined as:

$$\mathbf{S}_\phi(x_{t_i}, t_i, t_{i-1}) = \tfrac{t_i - t_{i-1}}{t_i}\big(\mathbf{g}_\phi(x_{t_i}, t_i) - x_{t_i}\big) + x_{t_i}$$

There are many different types of $\mathbf{S}_\phi$, e.g., the Heun solver introduced in EDM (Karras et al., 2022), or other high-order

solvers (Lu et al., 2022a;b; Zhou et al., 2024b; Xue et al., 2024). We use the Euler solver as an example here. With this solver, we can sample iteratively to get the final results.

### 2.2. Diffusion Distillation

A diffusion distillation method typically consists of a series of teacher models $\mathbf{H}$ and student models $\mathbf{F}_\theta$. These teacher models often have more steps than the student models. For example, in Progressive Distillation (PD) (Salimans & Ho, 2022), teacher models are defined as:

$$\mathbf{H} = \begin{cases} \mathbf{S}_\phi(\mathbf{S}_\phi((x_{t_i}, t_i, t_j), t_j, t_k), & i-j = j-k = 1 \\ \mathbf{F}_\theta^{sg}(\mathbf{F}_\theta^{sg}((x_{t_i}, t_i, t_j), t_j, t_k), & i-j = j-k \neq 1 \end{cases}$$

where $sg$ is the stop gradient to prevent gradient leakage to the teacher models.

Recently, an increasing number of distillation methods have found that adding an extra GAN loss alongside the distillation loss can be highly effective (Sauer et al., 2023; Xu et al., 2023; Kang et al., 2024; Sauer et al., 2024; Li et al., 2024a; Kim et al., 2024a). This observation motivates us to explore the mechanism underlying the performance gap between the original distillation methods and the gain provided by a simple GAN loss.

### 2.3. Inconsistent Local Minima

In these distillation methods, a teacher model $\mathbf{H}$ must pass through the neural network multiple times and contain more parameters than the student, while the student model $\mathbf{F}_\theta$ passes through the neural network in a single iteration and has fewer parameters. We reveal the potential problem

introduced by such a setup.

The difference between the teacher models and the student models leads to a significant inductive bias: the teacher model can transform between pixel space and latent space several times while the student model can only perform the transformation once. We speculate that this may produce different optimization landscapes and different local minima between the teacher and student model. Forcing the student model to approximate the teacher model's results may, therefore, yield sub-optimal results.

To verify our hypothesis, we use the Fréchet Inception Distance (FID) (Heusel et al., 2018) to compare two image sets. Specifically, we compute the FID between the teacher and student models, as well as the FID against the training-set images. FIDs are calculated using 50,000 images generated by each model with fixed seeds ranging from 0 to 49,999, and features for FID computation are extracted using an Inception model.

We define a series of teacher models with 2, 4, 6, 8, and 10 steps in a progressive distillation (PD) manner. For instance, a two-step teacher model is parameterized as:

$$H = F_\theta(F_\theta(x_{t_2}, t_2, t_1), t_1, t_0),$$

where $t_i \in \{T(2, 2), T(1, 2), T(0, 2)\}$, and similarly for other configurations. Both teacher and student models are trained using only the GAN loss and initialized from a well-trained score model in EDM (Karras et al., 2022). All teacher models achieve FID $\leq 2.2$ within a few training steps (less than 5 million training images), ensuring comparable performance across models. Finally, we compute the FID between the student model and each teacher model, following the evaluation protocol in EDM (Karras et al., 2022).

Figure 2**A.** illustrates the FIDs for teacher models with varying numbers of steps. While the FID values computed with the target dataset are similar across teacher models, those with more steps show higher FID values when compared with the one-step student model. This suggests a greater divergence from the student model as the number of teacher steps increases. Additionally, when we fix the teacher model's steps to two and vary the sigma of the intermediate time step, the FID computed with the one-step student decreases (Figure 2**B.**) Teacher models with higher sigma values generate images that are more similar to those produced by the one-step student model. More details of the loss function, time scheduler, and student/teacher model in distillation methods can be found in Appendix B.

These results reveal that the teacher and student models may achieve similar performance, but in different ways. Teachers with fewer steps are more similar to the one-step student, but there is still a significant difference between

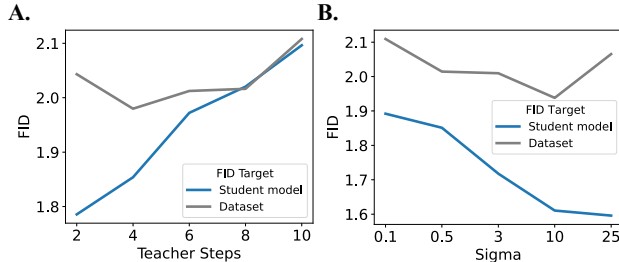

**A.** **B.**

*Figure 2.* The FIDs of teacher models. The FIDs are similar when computed against the target dataset across different teacher steps (gray in **A**) or sigmas of the intermediate steps (gray in **B**). When computed against the student model, the FID increases as the steps grow (blue in **A**) and decreases as the sigma decreases (steps=2, blue in **B**).

them (FID=1.78 between a two-step teacher and a one-step student). The student model fails to mimic the teacher model at the instance level, which might be the reason that limits the performance of distillation methods.

## 3. Fast Converge With An Exclusive GAN Objective

### 3.1. Training Without Distillation Objective

We have demonstrated that the key issue with distillation methods arises from the fact that the teacher and student models, despite sharing the same architecture, are optimized differently due to the teacher's multi-step process. Directly aligning the student's output with the teacher's at each instance, which forces the student into the teacher's specific local minima (e.g., requiring $x_{student}$ to match $x_{teacher}$), may be suboptimal for the student model. A simple GAN objective can bypass this issue, guiding the student to align with the overall data distribution rather than replicating precise instances, allowing it to refine $x_{student}$ even when it differs from $x_{teacher}$, and thus find an optimal solution in its own parameter space. In addition, using the real images instead of the teacher model's results (Lin et al., 2024) in GAN objective pushes the student toward the real data distribution, thus bypassing imperfections in the teacher's distribution and achieving better performance.

### 3.2. Baseline

Based on this reasoning, we create the D2O (Diffusion to One-Step) model based on a diffusion U-Net architecture with only a GAN objective. We first test its performance in a small-scale experiment on CIFAR-10 (Krizhevsky, 2009) as the baseline. The non-saturating GAN objective is used

*Table 1.* Discriminators that are either too strict or too weak can lead to model collapse. Multi-scale discriminators (LPIPS and PG) show better performance.

| Discriminator | FID ($\downarrow$) |
|---|---|
| StyleGAN2 (Scratch) | collapse |
| StyleGAN2 (Pre-trained) | collapse |
| VGG16 | 4.04 |
| LPIPS | 3.49 |
| PG | **2.21** |

*Table 2.* Augmentation leads to poor results. R1 regularization improves performance significantly.

| Augmentation | $\gamma_{r1}$ | FID ($\downarrow$) |
|---|---|---|
| ✓ | — | 2.21 |
| ✗ | — | 1.98 |
| ✗ | 1e-3 | 1.77 |
| ✗ | 1e-4 | **1.66** |
| ✗ | 1e-5 | 1.66 |

for the discriminator and the generator, respectively:

$$\max_{\mathbf{D}} \ \mathbb{E}_{\mathbf{x}}[\log(\mathbf{D}(\mathbf{x}))] + \mathbb{E}_{\mathbf{z}}[1 - \log(\mathbf{D}(\mathbf{G}_\theta(\mathbf{z})))]$$

$$\min_{\mathbf{G}_\theta} \mathbb{E}_{\mathbf{z}}[-\log(\mathbf{D}(\mathbf{G}_\theta(\mathbf{z})))]$$

With a pre-trained diffusion U-Net model, the score model $\mathbf{g}_\phi$ is used as the generator $\mathbf{G}_\theta$, and a pre-trained VGG16 (Simonyan & Zisserman, 2015) is used as a discriminator by default. EDM with NCSN++ (Song et al., 2021) architecture is used as the generator. We choose VGG16 (Simonyan & Zisserman, 2015) as a baseline discriminator. Adaptive augmentation (ADA) (Karras et al., 2020a) is adopted with r1 regularization (Mescheder et al., 2018) where $\gamma_{r1} = 0.01$ as suggested in StyleGAN2-ADA (Karras et al., 2020a). Our baseline setting shows good performance on the CIFAR-10 dataset with only 5M training images and already achieves performance (FID=4.04) close to that of the Consistency Distillation (Song et al., 2023) (FID=3.55) .

### 3.3. Discriminator

An appropriate discriminator is crucial since it represents the overall target distribution and determines the performance of the generator. The StyleGAN2 (both scratch and pre-trained with GAN objective) (Karras et al., 2020b), VGG, LPIPS (Zhang et al., 2018) , and Projected GAN (PG) (Sauer et al., 2022) discriminators are compared. Adaptive augmentation (ADA) is adopted for StyleGAN2, VGG, and LPIPS discriminators with r1 regularization where $\gamma_{r1} = 0.01$.

For the PG discriminator, we use a non-saturating version of PG objective:

$$\max_{\{\mathbf{D}_l\}} \sum_{l \in \mathcal{L}} \Big( \mathbb{E}_{\mathbf{x}}[\log \mathbf{D}_l(\mathbf{P}_l(\mathbf{x}))] + \mathbb{E}_{\mathbf{z}}[1 - \log(\mathbf{D}_l(\mathbf{P}_l(\mathbf{G}_\theta(\mathbf{z}))))] \Big)$$

$$\min_{\mathbf{G}_\theta} \sum_{l \in \mathcal{L}} \Big( \mathbb{E}_{\mathbf{z}}[-\log(\mathbf{D}_l(\mathbf{P}_l(\mathbf{G}_\theta(\mathbf{z}))))] \Big)$$

We use VGG16 with batch normalization (VGG16-BN) (Simonyan & Zisserman, 2015; Ioffe & Szegedy, 2015) and EfficientNet-lite0 (Tan & Le, 2020) as feature networks $\mathbf{P}$. We adopt differentiable augmentation (diffAUG) (Zhao et al., 2020) without a gradient penalty by default, following

the original work.

As illustrated in Table 1, discriminators with multiple scales are better than the vanilla VGG discriminator. Overall, the PG discriminator with a fusion feature based on VGG16-BN and EfficientNet-lite0 achieves the best results. Therefore, we will use the PG discriminator and objective in the following experiments.

### 3.4. Augmentation and Regularization

Differentiable augmentation (diffAUG) was introduced in (Zhao et al., 2020) to prevent the overfitting of discriminators, but we find that it leads to poor results in our method (Table 2). This may be because EDMs are pre-trained with data augmentation. We disable all augmentation in all of our further experiments.

In the original PG, regularization for the discriminator was not used. However, our experiments show that it is important to apply r1 regularization on the discriminator to prevent overfitting and it improves performance significantly (Table 2). The value of $\gamma_{r1}$ should be kept small since the discriminator of the Projected GAN generates multiple output logits, and a large $\gamma_{r1}$ may lead to a numerical explosion. With optimal settings, this setup can obtain satisfying results with only 0.2 million training images (FID=3.62). With 5 million training images, the results are close to SOTA results (FID=1.66).

## 4. Diffusion Objective as Generative Pre-training

### 4.1. Generative Pre-training

D2O attains strong results using far fewer training images than most previous distillation methods (See Appendix F for more detail). These gains are unlikely to come from the brief GAN fine-tuning alone, since 0.2 million images are insufficient for the model to learn a complex distribution from scratch. GAN training typically requires tens of millions of images, and training diffusion models often needs hundreds of millions or even billions of images.

*Table 3.* D2O-F: ablation experiments. ✓ indicates the corresponding layer can be tuned and ✗ indicates frozen layers. The percentage in the brackets is the proportion of the total parameters in these layers.

| Norm(7.9%) | Conv (85.8%) | QKV (2.1%) | Skip(4.0%) | FID ($\downarrow$) |
|:---:|:---:|:---:|:---:|:---:|
| ✓ | ✓ | ✓ | ✓ | 1.66 |
| ✓ | ✗ | ✓ | ✓ | **1.54** |
| ✓ | ✗ | ✗ | ✓ | 1.60 |
| ✓ | ✗ | ✗ | ✗ | 2.51 |

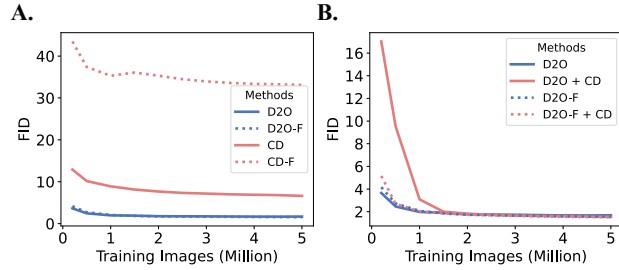

*Figure 3.* **A.** Performance of the D2O and CD models with (D2O-F and CD-F) and without (D2O and CD) freezing the convolutional layers. **B.** Effects of adding an extra CD loss to the D2O and D2O-F models.

Therefore, we speculate that the diffusion U-Net acquires an *intrinsic generative capability* during its original diffusion training, which the GAN fine-tuning phase merely unlocks. While directly using the pre-trained diffusion model for inference may be inefficient (Ma et al., 2023; Li et al., 2023a; Zou et al., 2024; Li et al., 2024b; Zhao et al., 2024; Chen et al., 2024a; Kim et al., 2024b; Chen et al., 2024b), it can be fine-tuned with a more direct and effective generative objective, such as a GAN-based objective, to utilize these intrinsic generative capabilities and produce a fast and efficient generative model.

This approach avoids redundancy by eliminating the role of noise as a redundant information destroyer during the iterative inference process. Instead, the noise serves as an index that maps a sample from the perturbed distribution onto its counterpart in the target distribution, similar to Consistency Models (Song et al., 2023; Song & Dhariwal, 2023) and GANs.

### 4.2. Achieving One-Step Generation via Freezing

To verify the feasibility, efficiency, and generality of this perspective, particularly in fine-tuning generative pre-training models for one-step generation, we create the D2O-F (Diffusion to One-Step Generators with Freezing) model in which most of the diffusion model's parameters are frozen during the fine-tuning with a GAN objective. If the model indeed relies on its pre-trained generative capabilities, we should expect to observe a performance similar to D2O.

To find the optimal setting, we conduct the ablation experiments in which different sets of parameters are frozen or tunable during training (Table 3). The results indicate that freezing most of the convolutional layers leads to the best performance. Further freezing the QKV projections (Vaswani et al., 2023) causes a slight decrease in the performance, and freezing the skip layers on the residual connections leads to worse results, suggesting that tuning these two types of layers is necessary.

Based on these results, the D2O model is trained with most of the original parameters locked (85.8% of the parameters contained in the convolutional layers) in the pre-trained

diffusion U-Net. D2O-F produces satisfying images with as few as 0.2 million training steps (FID=4.12). The performance further reaches near the SOTA level with only 5 million steps (FID=1.54). In comparison, training a generative model with similar performance typically requires tens or hundreds of millions of training steps (100M for StyleGAN2-ADA, 200M for EDM, on CIFAR-10). These results strongly support our theory that the initial diffusion training already provides a sufficient generative capability that can be rapidly fine-tuned for one-step generation.

Notably, with the majority of parameters in both the discriminator and the generator frozen, the training process of D2O-F is stable with minimal instances of mode collapse. Thereby, the freezing method circumvents the inherent instability of using GANs.

Finally, freezing parameters is different from lora-based methods (Luo et al., 2023; Lin et al., 2024). Lora-based methods adjust the convolutional parameters, although in a low-dimensional manner. By freezing the convolution layer, our method suggests that the diffusion models already possess the capacity for one-step generation.

### 4.3. Freezing Is Not Suitable for Distillation

The freezing method does not work well for the distillation methods. To demonstrate this, we compare the original CD and the CD with frozen convolutional layers (CD-F) (Fig. 3**A.**). CD-F performs poorly, which suggests that forcing the student model to emulate the teacher model requires adjustments in the convolutional layers. This further supports our hypothesis that forcing the student model to replicate the teacher model's outputs at the instance level will lead to suboptimal results due to the mismatch between their optimal local minima. This explains the inefficiency of distillation methods as a fine-tuning target: they require the model to learn generative capabilities anew.

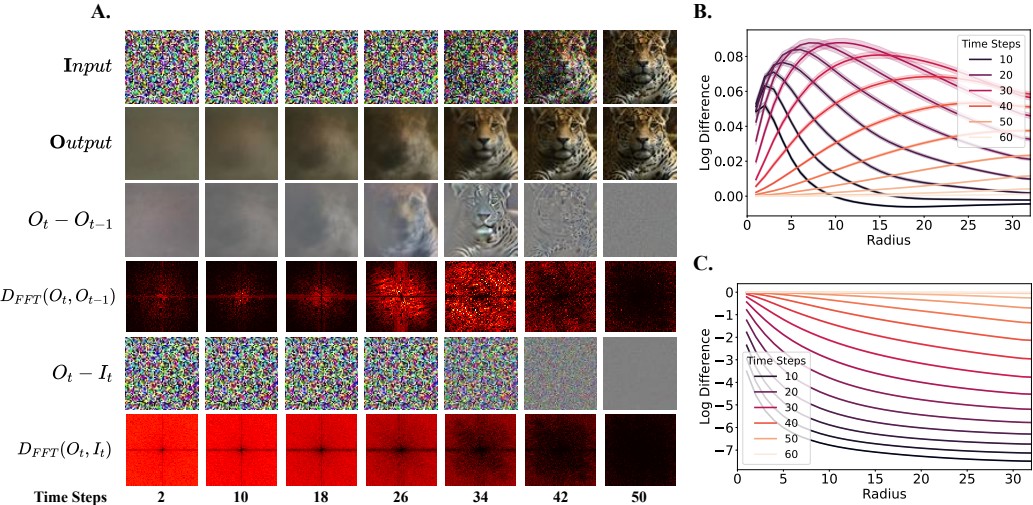

*Figure 4.* **A.** Spatial and frequency domain visualization. First row: Inputs at different time steps. Second row: Outputs at different time steps. Third and Fourth row: the difference between the current outputs and the previous outputs in the spatial domain (enhanced for clearer visualization) and the frequency domain. Fifth and Sixth row: the difference between the current outputs and the current inputs in the spatial and frequency domain. **B.** Radial Averaging of $D_{FFT}(O_t, O_{t-1})$, corresponding to the fourth row in **A**. Smaller radii indicate lower frequencies. During inference, the model selectively enhances increasingly higher frequency components at later time steps. **C.** Radial Averaging of $D_{FFT}(O_t, I_t)$, corresponding to the last row in **A**. All frequency bands are suppressed, which is more so in early time steps. That is because the model removes the redundant noise at each time step.

### 4.4. Extra Distillation Objective

Previous works (Sauer et al., 2023; Xu et al., 2023; Kang et al., 2024; Sauer et al., 2024; Li et al., 2024a; Kim et al., 2024a) employ both the distillation objective and the GAN objective. To assess their contribution to performance, we test our model with an additional distillation loss incorporated. We use the Consistency Distillation (CD) objective (Song et al., 2023) here, as it is easy to implement and performs well. Detailed information about the CD objective can be found in Appendix C. The comparisons are conducted on the CIFAR-10 dataset.

We find that the extra CD loss significantly slows down the convergence in the early stage. Although it performs slightly better than D2O, this advantage disappears when compared against D2O-F (Figure 3**B.**). In addition, adding CD loss nearly doubles the training resources and time.

Although the extra distillation loss may offer potential benefits, such as in image editing or classifier-free guidance (Ho & Salimans, 2022), these benefits may be achieved in our model with other methods such as adjusting the downstream task.

### 4.5. Frequency-Specific Processing

Though our experiments provide evidence for the general generative capabilities in diffusion models, the exact mechanism by which these capabilities emerge during training with the diffusion objective remains unclear. We speculate that these generative capabilities might involve frequency-specific processing.

We use the log-frequency difference as a core metric to quantify frequency response:

$$D_{FFT}(x_1, x_2) = \log\left(|FFT(x_1)| + 1\right) - \log\left(|FFT(x_2)| + 1\right),$$

where $x_1$ and $x_2$ represent two quantities to be compared.

**Global Frequency Evolution** We first notice that the information across the entire frequency range changes between the input and the output at different time steps, measured by $D_{FFT}(O_t, I_t)$ (Figure 4 **A.** and **C.**). This is consistent with the denoising process carried out by the model, which suppresses noise across the entire spectrum. Furthermore, when examining the frequency differences between the outputs of two consecutive time steps $D_{FFT}(O_t, O_{t-1})$, we observe a distinct pattern: low-frequency components are enhanced in the earlier time steps, while the enhanced frequencies shift toward the middle and high ranges in later time steps (Figure 4 **A.** and **B.**). This pattern provides quantitative support for the commonly observed phenomenon that diffusion inference progressively restores signals from low to high frequencies (Rissanen et al., 2023; Dieleman, 2023). It suggests that diffusion models may process different frequency components in a time-step-dependent manner during inference.

*Table 4.* A comprehensive comparison between D2O, D2O-F, and previous models on CIFAR-10.

| Method | NFE (↓) | Unconditional FID (↓) | Unconditional IS (↑) | Conditional FID (↓) |
|---|---|---|---|---|
| **Training From Scratch** | | | | |
| DDPM (Ho et al., 2020) | 1000 | 3.17 | | |
| DDIM (Song et al., 2022) | 100 | 4.16 | | |
| Score SDE (Song et al., 2021) | 2000 | | | 2.20 |
| DPM-Solve-3 (Lu et al., 2022a) | 48 | | | 2.65 |
| EDM (Karras et al., 2022) | 35 | 1.98 | | 1.79 |
| BigGAN (Brock et al., 2019) | 1 | | | 14.73 |
| StyleGAN2-ADA (Karras et al., 2020a) | 1 | 2.92 | 9.82 | 2.42 |
| SAN (Takida et al., 2024) | 1 | **1.36** | | |
| iCT (Song & Dhariwal, 2023) | 1 | 2.83 | 9.54 | |
| iCT (Song & Dhariwal, 2023) | 2 | 2.46 | 9.80 | |
| iCT-deep (Song & Dhariwal, 2023) | 1 | 2.51 | 9.76 | |
| iCT-deep (Song & Dhariwal, 2023) | 2 | 2.24 | 9.89 | |
| **Post-training** | | | | |
| PD (Salimans & Ho, 2022) | 1 | 9.12 | | |
| DFNO (Zheng et al., 2023) | 1 | 3.78 | | |
| CD (Song et al., 2023) | 1 | 3.55 | | |
| CD (Song et al., 2023) | 2 | 2.93 | | |
| CTM (Kim et al., 2024a) | 1 | 1.98 | | 1.73 |
| CTM (Kim et al., 2024a) | 2 | 1.87 | | 1.63 |
| DMD (Yin et al., 2023) | 1 | 2.62 | | |
| SiD (Zhou et al., 2024a), $\alpha = 1.0$ | 1 | 2.02 | 10.01 | 1.93 |
| SiD (Zhou et al., 2024a), $\alpha = 1.2$ | 1 | 1.92 | 9.98 | 1.71 |
| D2O (ours) | 1 | 1.66 | **10.11** | 1.58 |
| D2O-F (ours) | 1 | 1.54 | 10.10 | **1.44** |

*Table 5.* A comprehensive comparison between D2O, D2O-F, and previous work on AFHQv2 64x64 and FFHQ 64x64.

| Method | NFE (↓) | FID (↓) AFHQv2 | FID (↓) FFHQ |
|---|---|---|---|
| **Training From Scratch** | | | |
| EDM (Karras et al., 2022) | 79 | 1.96 | 2.39 |
| **Post-training** | | | |
| BOOT (Gu et al., 2023) | 1 | | 9.0 |
| SiD (Zhou et al., 2024a), $\alpha = 1.0$ | 1 | 1.62 | 1.71 |
| SiD (Zhou et al., 2024a), $\alpha = 1.2$ | 1 | 1.71 | 1.55 |
| D2O (ours) | 1 | **1.23** | 1.08 |
| D2O-F (ours) | 1 | 1.31 | **0.85** |

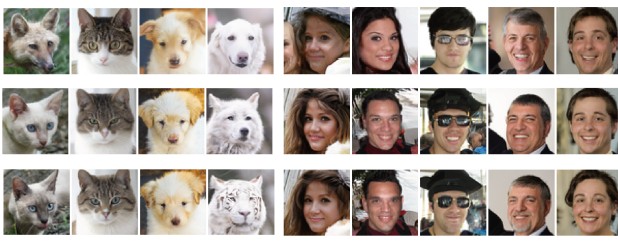

*Figure 5.* Sample comparison between EDM (Top), D2O (Middle) and D2O-F (Bottom) on AFHQv2 (left) and FFHQ (right).

**Block-wise Frequency Specialization** We also examine the frequency response of the U-Net blocks. Our analyses reveal a natural frequency-specific processing pattern, influenced by the diffusion U-Net structure (Figure 6). The deep blocks, operating at lower resolutions, are limited in their ability to process high-frequency information. In contrast, the shallow blocks, operating at higher resolutions, are better suited for capturing a broader range of high-frequency information. In addition, there is evidence of frequency-specific processing across different blocks at various time steps. High-frequency components often undergo more significant changes in the shallower layers, particularly during later time steps (See Appendix E for more details).

These results hint that frequency-specific processing may underlie the generative capabilities observed in diffusion models, which could be rapidly adapted and integrated with a simple GAN objective to efficiently produce a one-step generator.

# 5. Comprehensive Performance Test

## 5.1. Basic Setting

Finally, we conduct a comprehensive evaluation of both D2O and D2O-F on CIFAR-10, AFHQv2 64x64 (Choi et al., 2020), FFHQ 64x64 (Karras et al., 2019), and ImageNet 64x64 (Deng et al., 2009). The results for class-conditional generation and the comparison against previous studies are reported on CIFAR-10 and ImageNet 64x64. The pre-trained diffusion models are from EDM. We report FID for all datasets, Inception Score (IS) (Salimans et al., 2016) for CIFAR-10, precision and recall metric (Kynkäänniemi et al., 2019) for ImageNet 64x64 following previous works. For the discriminator, we use VGG16 with batch normalization and EfficientNet-lite0 as the feature network for CIFAR-10, DeiT (Touvron et al., 2021) , and EfficientNet-lite0 for the other datasets. The optimal settings for discriminator, augmentation, and regularization determined in previous experiments are used. More implementation details can be found in Appendix C.

## 5.2. Results

The results can be found in Table 4, 5 , and 6. Both D2O and D2O-F achieve competitive results on all datasets tested. Our methods not only are suitable for unconditional generation but also show great performance on class-conditional generation tasks. Moreover, D2O-F outperforms D2O by a significant margin in most datasets, except AFHQv2 64x64. On CIFAR-10, our methods achieve an FID of 1.54 and an IS of 10.11. Similarly, superior performance is found on AFHQv2 64x64 (FID=1.23) and FFHQ 64x64 (FID=0.85). In ImageNet 64x64, the model also achieves SOTA performance (FID=1.16, precision=0.77).

We further compare the outputs from the converted one-step

*Table 6.* A comprehensive comparison between D2O, D2O-F, and the competing models on ImageNet 64x64 (class-conditional).

| Method | NFE (↓) | FID (↓) | Prec. (↑) | Rec. (↑) |
|---|---|---|---|---|
| **Training From Scratch** | | | | |
| RIN (Jabri et al., 2023) | 1000 | 1.23 | | |
| DDPM (Ho et al., 2020) | 250 | 11.00 | 0.67 | 0.58 |
| ADM (Dhariwal & Nichol, 2021) | 250 | 2.07 | 0.74 | 0.63 |
| EDM (Karras et al., 2022) | 79 | 2.64 | | |
| iCT (Song & Dhariwal, 2023) | 1 | 4.02 | 0.70 | 0.63 |
| iCT (Song & Dhariwal, 2023) | 2 | 3.20 | 0.73 | 0.63 |
| iCT-deep (Song & Dhariwal, 2023) | 1 | 3.25 | 0.72 | 0.63 |
| iCT-deep (Song & Dhariwal, 2023) | 2 | 2.77 | 0.74 | 0.62 |
| BigGAN-deep (Brock et al., 2019) | 1 | 4.06 | 0.79 | 0.48 |
| StyleGAN2-XL (Sauer et al., 2022) | 1 | 1.51 | | |
| **Post-training** | | | | |
| PD (Salimans & Ho, 2022) | 1 | 15.39 | | |
| BOOT (Gu et al., 2023) | 1 | 16.3 | 0.68 | 0.36 |
| DFNO (Zheng et al., 2023) | 1 | 7.83 | | 0.61 |
| CD (Song et al., 2023) | 1 | 6.20 | 0.68 | 0.63 |
| CD (Song et al., 2023) | 2 | 4.70 | 0.69 | **0.64** |
| CTM (Kim et al., 2024a) | 1 | 1.92 | 0.70 | 0.57 |
| CTM (Kim et al., 2024a) | 2 | 1.73 | 0.64 | 0.57 |
| DMD (Yin et al., 2023) | 1 | 2.62 | | |
| SiD (Zhou et al., 2024a), $\alpha$=1.0 | 1 | 2.02 | 0.73 | 0.63 |
| SiD (Zhou et al., 2024a), $\alpha$=1.2 | 1 | 1.52 | 0.74 | 0.63 |
| sCD-S (Lu & Song, 2025) | 1 | 2.97 | | |
| sCD-S (Lu & Song, 2025) | 2 | 2.07 | | |
| ECM-S (Geng et al., 2024) | 1 | 4.05 | | |
| ECM-S (Geng et al., 2024) | 2 | 2.79 | | |
| DMD2 (Yin et al., 2024) | 1 | 1.51 | | |
| DMD2 (Yin et al., 2024), longer training | 1 | 1.26 | | |
| MSD (Song et al., 2024), 4 students, DM | 1 | 2.37 | | |
| MSD (Song et al., 2024), 4 students, ADM | 1 | 1.20 | | |
| D2O(ours) | 1 | 1.42 | **0.77** | 0.59 |
| D2O-F(ours) | 1 | **1.16** | 0.75 | 0.60 |

generator and the original diffusion model EDM (Figure 5, more samples can be found in Appendix G). The produced images are similar, and there is a certain degree of variation, as we do not force the student model to replicate the teacher model at the instance level, allowing it to perform differently to surpass the original model. Based on FID (calculated using EDM's results on CIFAR-10), D2O-F (1.87) performs similarly to the other distillation methods like CD (1.76) and significantly better than StyleGAN-XL (2.60). Together, these results demonstrate the robustness and effectiveness of our methods, supporting our hypothesis that one-step generation utilizes the generative capabilities gained from pre-training.

## 6. Discussion

### 6.1. Generative Pre-training

Here, we propose that diffusion model training can be viewed as a generative pre-training process. With this view, the generative pre-training should consist of two key processes: a destruction process that reduces information to a known noise distribution and a self-supervised training process that trains the model to recover samples with varying

noise levels. The training process helps the model learn the necessary generative capabilities. The destruction process allows the model to sample from a known distribution and generate or edit images according to the task.

An architecture based on discrete tokens, such as autoregressive models, could also be framed within this framework with a few simple adjustments. Specifically, instead of merely removing tokens, we could replace them with tokens randomly sampled from a discrete uniform distribution (mixing these tokens is also a viable option). After the pre-training process, we can fine-tune the pre-trained model by sampling a set of random tokens and adopting a simple GAN objective to achieve one-step generation.

### 6.2. Limitation and Future Work

Our model is based on the diffusion U-Net architecture. The effectiveness of our approach on different architectures, such as DiT (Peebles & Xie, 2023), remains to be explored. Additionally, while the proposed generative pre-training method demonstrates good performance on the one-step generation task across several datasets, including the relatively complex ImageNet 64x64, it has yet to be tested on datasets with higher resolution and more complex generation tasks, such as the COCO (Lin et al., 2015) dataset. Finally, our efforts to explore the generative capabilities through the diffusion models' frequency response are only preliminary and need to be expanded. In the future, we will focus on extending our methods to more architectures, datasets, and tasks, as well as exploring the underlying mechanisms of how the diffusion objective provides universal capacities to diffusion models.

## 7. Conclusion

In this work, we aim to improve the efficiency of training one-step generative models from diffusion models. We first identify a fundamental limitation of distillation methods: the teacher and student models may converge to distinct local minima and the discrepancy hinders knowledge transfer. Therefore, we create the D2O model with a GAN objective to circumvent this issue. The D2O model produced results beyond our expectations, and we further explored the underlying mechanism. We propose that the training of the original model with the diffusion objective can be viewed as generative pre-training, where the model learns generalizable capabilities that can be rapidly fine-tuned and integrated into a one-step generator. To demonstrate the feasibility of the idea, we build the D2O-F model in which a significant portion of the original model parameters are frozen during fine-tuning. The model achieves competitive performance with far fewer training images, supporting our hypothesis that the one-step generation can be achieved through rapid fine-tuning of the pre-trained generative capabilities in diffu-

sion models. The frequency analysis provides a preliminary explanation for the freezing technique by revealing the temporal and block-wise frequency specialization in diffusion models. Together, our work provides key intuition for developing efficient and high-performance generative models in the future.

## Impact Statement

The primary advantage of transforming diffusion models into one-step generators lies in increased efficiency, which reduces the computational resources needed for generation. This not only leads to energy savings but also makes high-performance models accessible to users with standard computing resources, promoting fairness in AI usage.

However, similar to other generative models, there are potential risks, such as the misuse of these models to generate harmful content like violence or pornography. These risks are already part of ongoing industry discussions, and responsible deployment, along with clear ethical guidelines, is essential to prevent misuse.

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

## A. Related Work

**Image Generation**   GAN (Goodfellow et al., 2014; Brock et al., 2019; Karras et al., 2020b) models have dominated the image generation field for a long time. Quantization-based generative models (Esser et al., 2021; Chang et al., 2022; Li et al., 2023b) first encode images into discrete tokens, then use a transformer to model the probability distribution between the tokens. Diffusion Models (Sohl-Dickstein et al., 2015; Song & Ermon, 2020a; Ho et al., 2020; Song & Ermon, 2020b; Song et al., 2021) or score-based generative models try to learn an accuracy estimation of scores (the gradient of the log probability density) to sample from a perturbed distribution with a Gaussian kernel to the image distribution. Diffusion models have achieved great success in image generation (Dhariwal & Nichol, 2021; Nichol et al., 2022; Ramesh et al., 2022; Saharia et al., 2022) and are widely used in different fields.

**Accelerating Diffusion Inference**   Many recent works tried to accelerate the inference process of diffusion models, often focusing on the redundancy inherent in these models. This typically involves both architectural and temporal redundancy (Ma et al., 2023; Li et al., 2023a; Zou et al., 2024; Li et al., 2024b; Zhao et al., 2024; Chen et al., 2024a; Kim et al., 2024b; Chen et al., 2024b). These methods often involve reusing feature maps or outputs from neighboring time steps to reduce the computational cost of inference.

**Diffusion Distillation**   One of the challenges of diffusion models in practice is the high computational cost incurred during fine multi-step generation. A series of distillation methods (Salimans & Ho, 2022; Gu et al., 2023; Song et al., 2023; Yin et al., 2023; Zhou et al., 2024a;c; Hsiao et al., 2024) have been proposed to distill diffusion models to one-step generators. Distillation models with GAN were introduced recently. GAN loss has been used as an auxiliary loss of distillation loss (Sauer et al., 2023; Xu et al., 2023; Kang et al., 2024; Sauer et al., 2024; Li et al., 2024a; Kim et al., 2024a), or a replacement of instance-level loss (L2 or LPIPS) as in (Lin et al., 2024). Both of these approaches achieved good performance, demonstrating the potential of GAN-based distillation.

**Frequency Perspective**   Recently, several studies investigated diffusion models from a frequency perspective. For example, in (Rissanen et al., 2023), the authors analyze the changes induced by the diffusion forward process on images from a frequency viewpoint. In (Lee et al., 2024), analysis was conducted based on the frequency differences between inputs and outputs. In (Wang et al., 2023; Huang et al., 2024), a frequency-based improvement method was proposed, enhancing the model's efficiency and performance.

## B. Definition of Typical Distillation Methods

Progressive Distillation (PD) uses a progressive distillation strategy. Using $\mathbf{S}_\phi$ with a sufficient number of steps, e.g., $N_{max} = 1024$, as a teacher, PD first distills a student model $\mathbf{F}_\theta$ with fewer steps, typically $N_1 = \frac{1}{2}N_{max} = 512$. Then with the fixed student model $\mathbf{F}_\theta^{sg}$ as teacher where $sg$ means stop gradient, PD further distills the student with $N_2 = \frac{1}{4}N_{max} = 256$. This progress is repeated until a one-step generator is produced.

Consistency Distillation (CD) and Consistency Trajectory Model (CTM) are similar. Both of these two methods adopt a joint training strategy in which student model $\mathbf{F}_\theta$ is used to be part of the teacher model $\mathbf{H}(x_t, t, u, s) = \mathbf{F}_\theta^{sg}(\mathbf{S}_\phi(x_t, t, u), u, s)$ at the same time. CTM uses an iterative solver $\mathbf{S}^*_\phi$ here and it further projects $v_s$ and $\bar{v}_s$ to the pixel space by $\mathbf{F}_\theta^{sg}(v_s, s, \sigma_{min})$ and $\mathbf{F}_\theta^{sg}(\bar{v}_s, s, \sigma_{min})$ and then calculates $\mathbf{D}(v_{\sigma_{min}}, \bar{v}_{\sigma_{min}})$ as loss. The three models are summarized in Table 7, 8 and 9. We modify the definition of PD with a EDM manner here to be consistent. $t, u, s$ are the start, intermediate and end time steps of the trajectory, respectively. PD sets the $u$ to the "middle" of $t$ and $s$. CD sets the $u$ as the successor of $t$ and fixes the $s$ to zeros. PD uses a separate training strategy since the teacher model is student model from previous training, and PD supervises $f_\theta$ with the intersection of $x = 0$ and $\text{Slope}(x_t, \bar{v}_s)$, where $\text{Slope}(x_t, \bar{v}_s)$ is the slope between $x_t$ and $\bar{v}_s$.

*Table 7.* Sampling strategies for time steps of commonly used distillation methods

| Method | t | u | s |
|--------|---|---|---|
| PD | $T(i, N), 2 \leq i \leq N$ | $T(\frac{1}{2}(i+j), N)$ | $T(j, N), 0 \leq j < (i-1)$ |
| CD | $T(i, N), 1 \leq i \leq N$ | $T(i-1, N)$ | $T(0, N)$ |
| CTM | $T(i, N), 2 \leq i \leq N$ | $T(j, N), 1 \leq j \leq i$ | $T(k, N), 0 \leq k \leq j$ |

*Table 8.* Loss function of commonly used distillation methods

| Method | Loss Function ($\mathcal{L}$) |
|--------|-------------------------------|
| PD | $\mathbf{D}(\mathbf{f}_\theta(x_t, t), \frac{t}{t-s}(\mathbf{H}(x_t, t, u, s) - x_t) + x_t)$ |
| CD | $\mathbf{D}(\mathbf{F}_\theta(x_t, t, s), \mathbf{H}(x_t, t, u, s))$ |
| CTM | $\mathbf{D}(\mathbf{F}_\theta^{sg}(\mathbf{F}_\theta(x_t, t, s), s, \sigma_{min}), \mathbf{F}_\theta^{sg}(\mathbf{H}(x_t, t, u, s), s, \sigma_{min}))$ |

*Table 9.* Definition of teacher and student model of commonly used distillation methods

| Method | Student Model ($\mathbf{F}_\theta$) | Teacher Model ($\mathbf{H}$) |
|--------|-------------------------------------|------------------------------|
| PD | $\frac{t-s}{t}(\mathbf{f}_\theta(x_t, t) - x_t) + x_t$ | $\begin{cases} \mathbf{S}_\phi(\mathbf{S}_\phi(x_t, t, u), u, s), & N_i = N_{max} \\ \mathbf{F}_\theta^{sg}(\mathbf{F}_\theta^{sg}(x_t, t, u), u, s), & N_i \neq N_{max} \end{cases}$ |
| CD | $\frac{t-s}{t}(\mathbf{f}_\theta(x_t, t) - x_t) + x_t$ | $\mathbf{F}_\theta^{sg}(\mathbf{S}_\phi(x_t, t, u), u, s)$ |
| CTM | $\mathbf{f}_\theta(x_t, t, s)$ | $\mathbf{F}_\theta^{sg}(\mathbf{S}_\phi^*(x_t, t, u), u, s)$ |

## C. Implementation Details

We use EDMs as the original models for all datasets and methods to ensure a fair comparison. EMA decay is applied to the weights of the generator for sampling following previous work. EMA decay rate is calculated by EMA Halflife and EMA warmup is used as in EDM. EMA warmup ratio is set to 0.05 for all datasets, and this leads to a gradually growing EMA decay rate. We use Adam optimizer with $\beta_1 = 0, \beta_2 = 0.99$ without weight decay for both the generator and the discriminator. No gradient clip is applied. For class-conditional generation, we use no classifier-guidance but simple class embedding in the discriminator following StyleGAN2-XL. Mixed precision training is adopted for all experiments with *BFloat16* data type, and results with *Float16* and *Float32* are similar. We apply no learning rate scheduler. Images are resized to 224 first before they are fed to the PG discriminator or CD-LPIPS loss. All models are trained on a cluster of NVIDIA A100 GPUs. Hyperparameters used for D2O and D2O-F training can be found in Table 10. D2O and D2O-F share the same hyperparameters except for the number of training images, since D2O-F will not overfit when D2O gets worse FID.

*Table 10.* Hyperparameters used for training D2O and D2O-F

| Hyperparameter | CIFAR-10 | AFHQ $64 \times 64$ | FFHQ $64 \times 64$ | ImageNet $64 \times 64$ |
|----------------|----------|----------|----------|----------|
| G Architecture | NSCN++ | NSCN++ | NSCN++ | ADM |
| G LR | 1e-4 | 2e-5 | 2e-5 | 8e-6 |
| D LR | 1e-4 | 4e-5 | 4e-5 | 4e-5 |
| Optimizer | Adam | Adam | Adam | Adam |
| $\beta_1$ of Optimizer | 0 | 0 | 0 | 0 |
| $\beta_2$ of Optimizer | 0.99 | 0.99 | 0.99 | 0.99 |
| Weight decay | No | No | No | No |
| Batch size | 256 | 256 | 256 | 512 |
| $\gamma_{r1}$ (Regularization) | 1e-4 | 1e-4 | 1e-4 | 4e-4 |
| EMA half-life (Mimg) | 0.5 | 0.5 | 0.5 | 50 |
| EMA warmup ratio | 0.05 | 0.05 | 0.05 | 0.05 |
| Training Images (Mimg) | 5 (D2O) 5 (D2O-F) | 5 (D2O) 10 (D2O-F) | 5 (D2O) 10 (D2O-F) | 5 (D2O) 5 (D2O-F) |
| Mixed-precision (BF16) | Yes | Yes | Yes | Yes |
| Dropout probability | 0.0 | 0.0 | 0.0 | 0.0 |
| Augmentations | No | No | No | No |
| Number of GPUs | 8 | 8 | 8 | 8 |

We follow the parameterization in Consistency Model for all of our models:

$$c_{\text{skip}}(t) = \frac{\sigma_{\text{data}}^2}{(t - \epsilon)^2 + \sigma_{\text{data}}^2}, \quad c_{\text{out}}(t) = \frac{\sigma_{\text{data}}(t - \epsilon)}{\sqrt{\sigma_{\text{data}}^2 + t^2}}$$

We set $t = \sigma_{max}$ when applying D2O/D2O-F with $c_{\text{skip}} \approx 0$ and $c_{\text{out}} \approx 1$. When training with CD loss, we use Heun solver. The time step scheduler is slightly different from the scheduler defined in EDM. We reverse the index and define it as a function of the current step and total steps:

$$T(i, N) = \left( \sigma_{max}^{\frac{1}{\rho}} + \left(1 - \frac{i}{N}\right) \left( \sigma_{min}^{\frac{1}{\rho}} - \sigma_{max}^{\frac{1}{\rho}} \right) \right)^{\rho}$$

Following Consistency Distillation, we set $N = 18$, $\rho = 7$, $\sigma_{min} = 0.002$ and $\sigma_{max} = 80$.

## D. CLIP-FID Results

Potential data leakage in FID when using a discriminator pre-trained on ImageNet has been a concern (Kynkäänniemi et al., 2023). We provide CLIP-FID in Table D. Our method consistently shows superior or competitive performance with significantly less training data. The result indicates that the superior performance is not due to information leakage but the pretrained ability gained in diffusion training.

*Table 11.* Performance comparison across datasets: CLIP-FID, FID, and training images

| Dataset | Model | CLIP-FID | FID | Training Images |
|---------|-------|----------|-----|-----------------|
| CIFAR10 | EDM | **0.53** | 1.98 | - |
| | CD | 1.26 | 4.10 | ~100M |
| | SiD | 0.65 | 1.92 | ~400M |
| | D2O-F | 0.66 | **1.56** | **~5M** |
| FFHQ | EDM | 1.18 | 2.39 | - |
| | SiD | **0.80** | 1.55 | ~500M |
| | D2O-F | 0.81 | **0.83** | **~9M** |
| AFHQv2 | EDM | 0.40 | 1.96 | - |
| | SiD | 0.32 | 1.62 | ~300M |
| | D2O-F | **0.18** | **1.24** | **~7M** |
| ImageNet | EDM | 0.82 | 2.64 | - |
| | CD | 2.93 | 6.87 | ~1000M |
| | SiD | 0.75 | 1.52 | ~930M |
| | D2O-F | **0.51** | **1.13** | **~6M** |

## E. Details of Block-Wise Frequency Analysis

We first notice a natural correlation between the U-Net architecture and spatial frequency (Figure 6**A.**). The deeper layers, which operate at lower resolutions, are limited in their ability to process high-frequency information. The shallower layers, operating at higher resolutions, are better equipped to capture a broader range of high-frequency information.

Next, we examine how frequency contributions evolve over time across layers and their alignment with the overall frequency response in the diffusion process. We calculate the frequency difference between the input and output of each layer at different time steps $D_t^b = D_{FFT}(x_t^b, x_t^{b-1})$, where $t$ refers to the time step and $b$ refers to the block index. There are significant specialization patterns of different blocks (Figure 6 **B.** and **C.**).

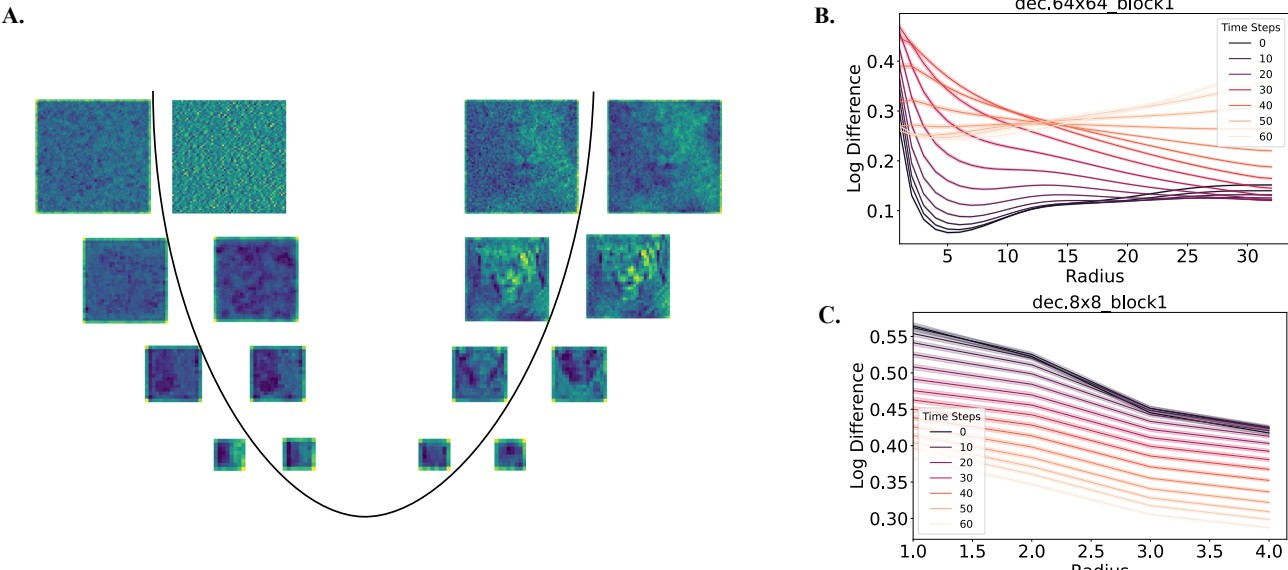

*Figure 6.* **A.** Average feature maps at different depths. The inner region corresponds to the skip path, while the outer region corresponds to the convolution path consisting of normalization, activation, and convolutional layers. From deeper to shallower layers, the decoder exhibits a transition from low to high-frequency contributions in the spatial domain. **B.** Radial averaging of the log difference between the output and input of a shallow layer reveals that the high-frequency component increases over time. **C.** Radial averaging of the log difference between the output and input of a deep layer. Due to the lower resolution, this layer exhibits a smaller radius and frequency range.

## F. Training Efficiency Comparison

We provide further training efficiency comparisons between our methods and the other approaches on ImageNet $64\times64$. The name in the quote is the pre-trained diffusion model. With similar teacher models (EDM and EDM2), our methods require significantly fewer training images while delivering better performance.

*Table 12.* Comparison of distilled one-step generators on ImageNet $64\times64$

| Method | FID ↓ | Training Images (M)↓ | Params (M) |
|---|---|---|---|
| BOOT (EDM) | 16.30 | 307 | 280 |
| DMD (EDM) | 2.62 | 117 | 280 |
| ECM-S (EDM2) | 5.51 | 12 | 280 |
| ECM-S, longer training (EDM2) | 4.05 | 102 | 280 |
| ECM-XL (EDM2) | 3.35 | 12 | 1119 |
| ECM-XL, longer training (EDM2) | 2.49 | 102 | 1119 |
| sCD-S (EDM2+TrigFlow) | 2.97 | 819 | 280 |
| sCD-XL (EDM2+TrigFlow) | 2.44 | 819 | 1119 |
| **D2O-F (EDM)** | **1.16** | **5** | 280 |

## G. Samples

We include EDM's results for comparison. The initial noises are the same as in all models. D2O and D2O-F generate images similar to the original model (EDM). But they are not identical, because our methods allow the student model to perform differently from the teacher model.

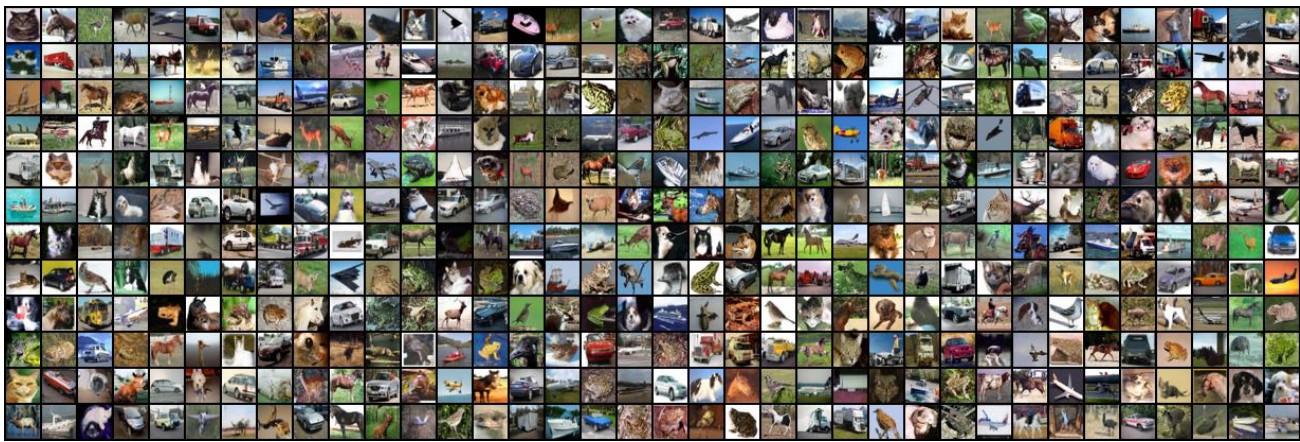

*Figure 7.* CIFAR-10, EDM, NFE=18, FID=1.96

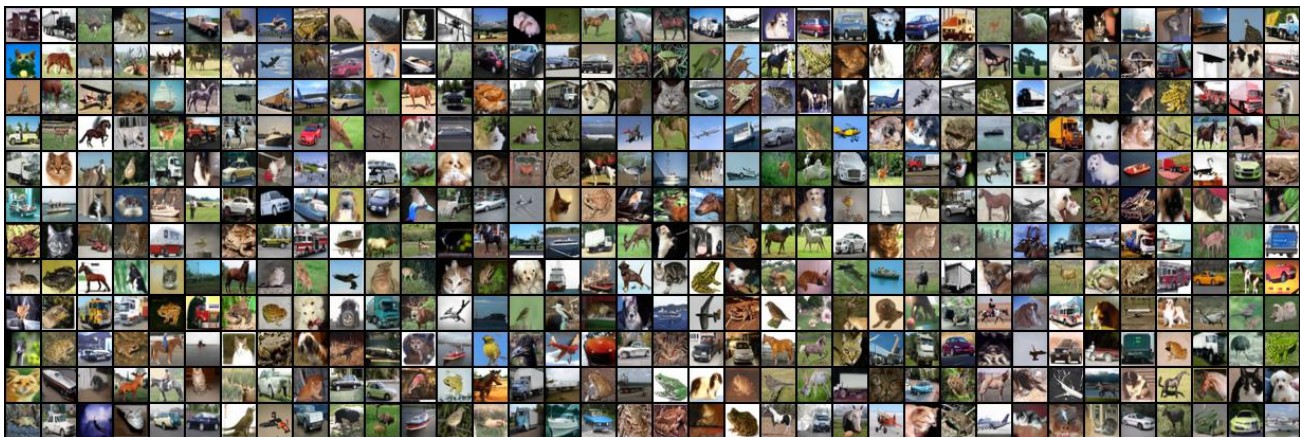

*Figure 8.* CIFAR-10, D2O, NFE=1, FID=1.66

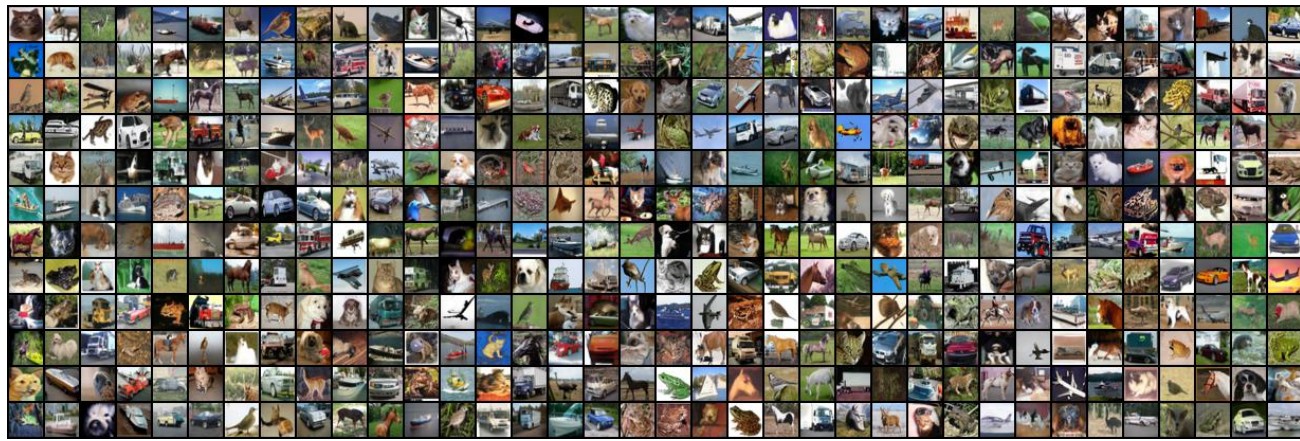

*Figure 9.* CIFAR-10, D2O-F, NFE=1, FID=1.54

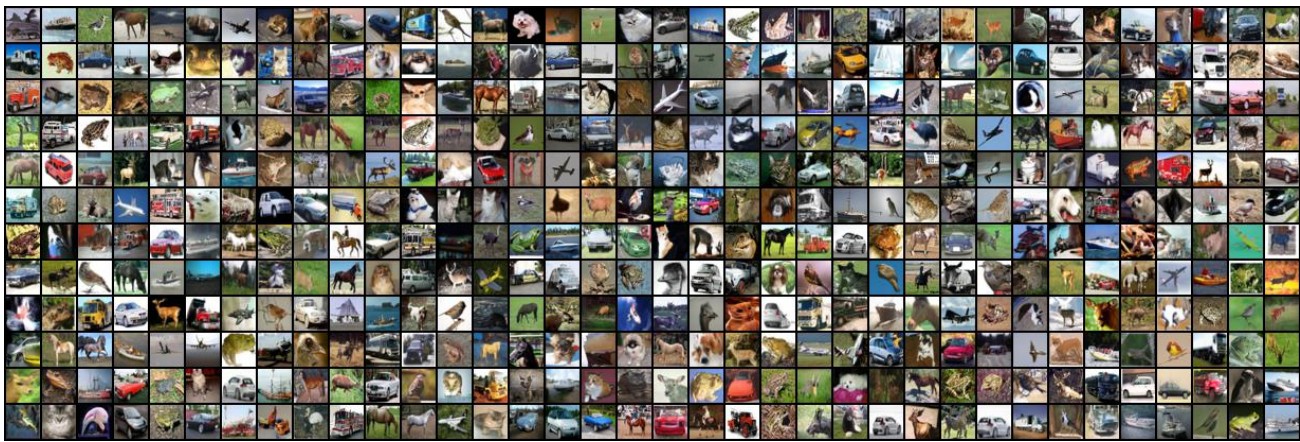

*Figure 10.* CIFAR-10 (conditional), EDM (VE), NFE=18, FID=1.82

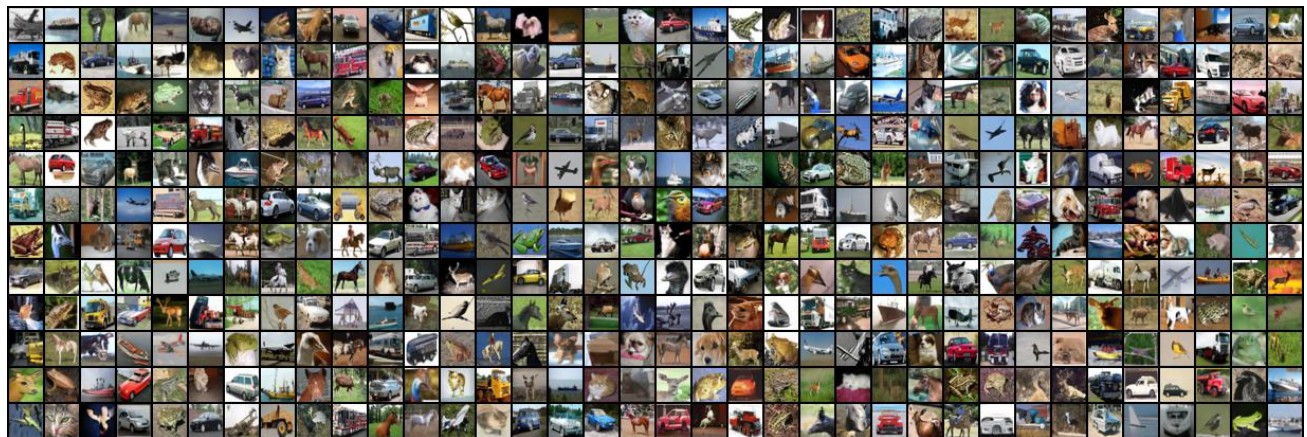

*Figure 11.* CIFAR-10 (conditional), D2O, NFE=1, FID=1.58

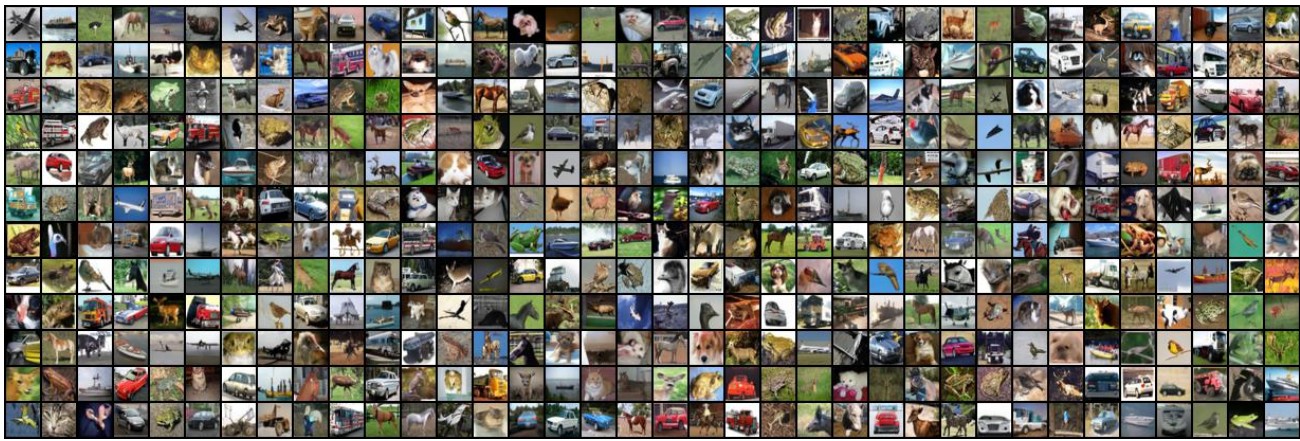

*Figure 12.* CIFAR-10 (conditional), D2O-F, NFE=1, FID=1.44

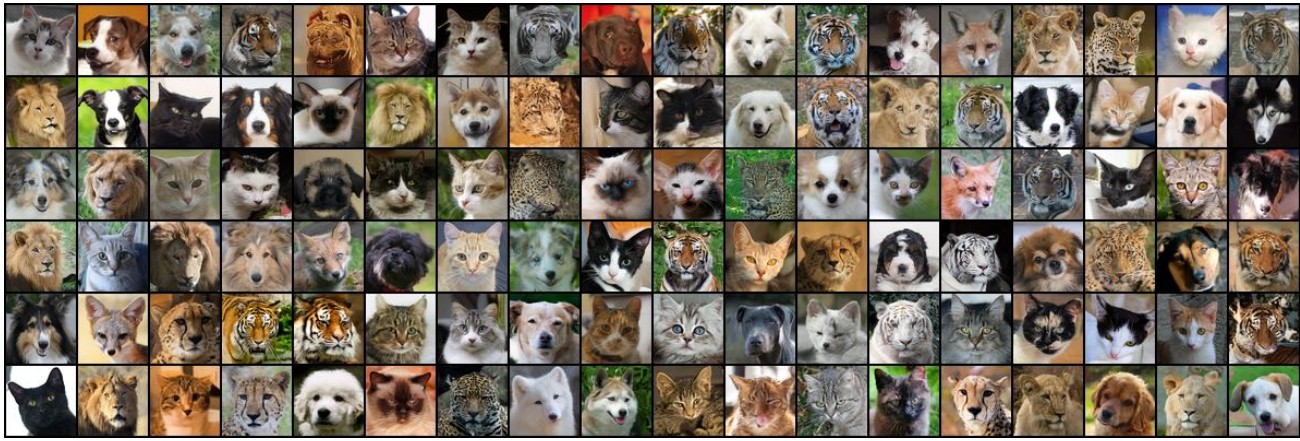

*Figure 13.* AFHQv2 64x64, VE, NFE=79, FID=2.17

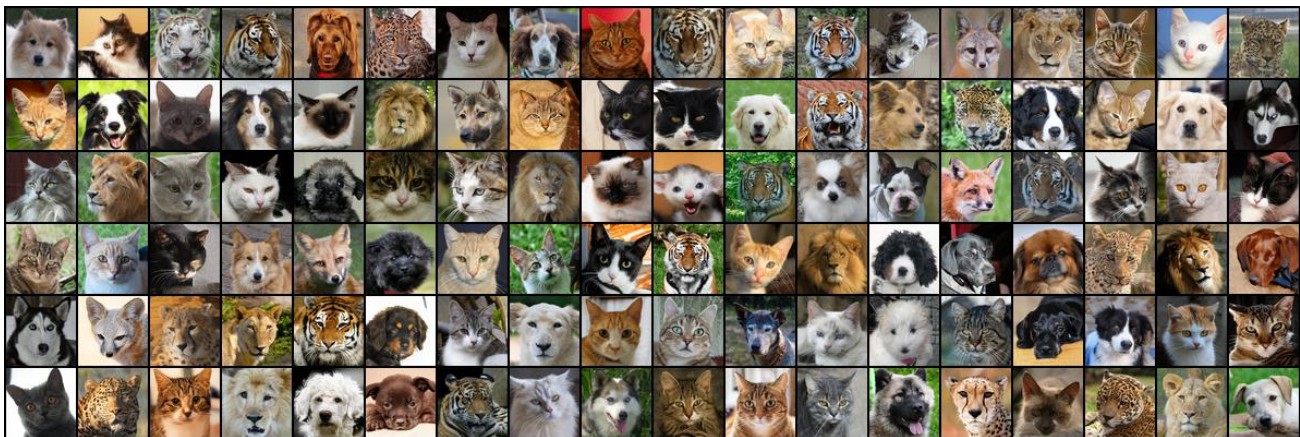

*Figure 14.* AFHQv2 64x64, D2O, NFE=1, FID=1.23

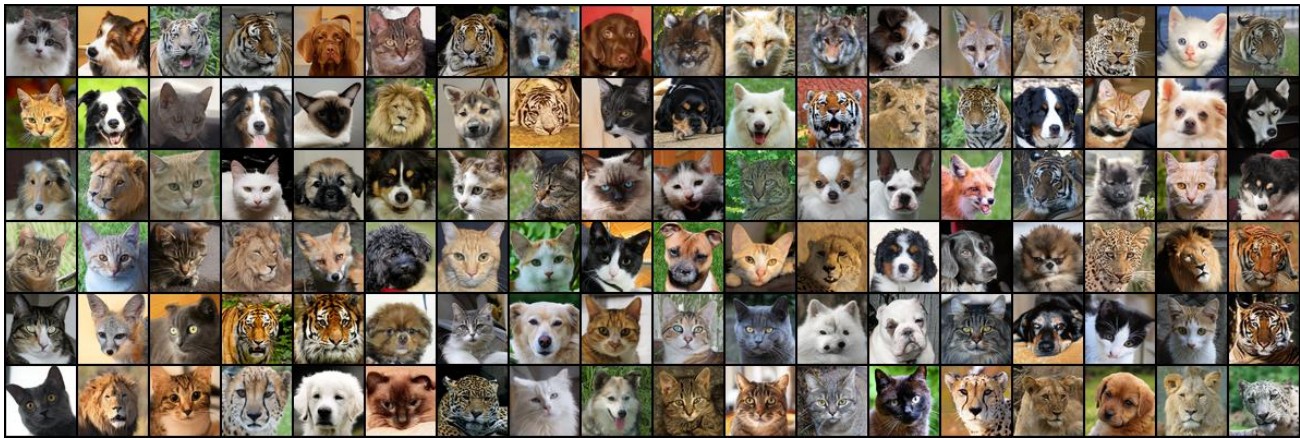

*Figure 15.* AFHQv2 64x64, D2O-F, NFE=1, FID=1.31

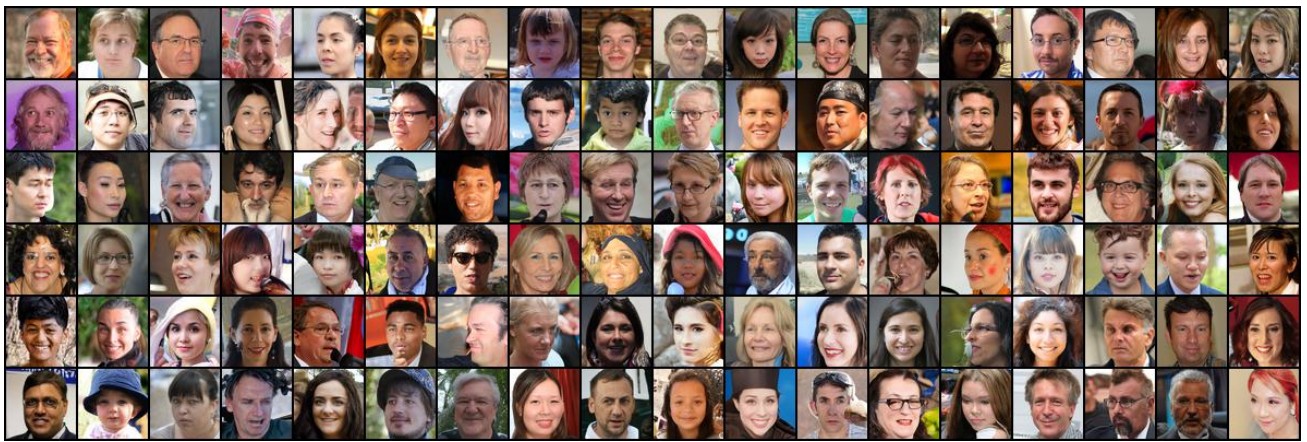

*Figure 16.* FFHQ 64x64, EDM (VE), NFE=79, FID=2.60

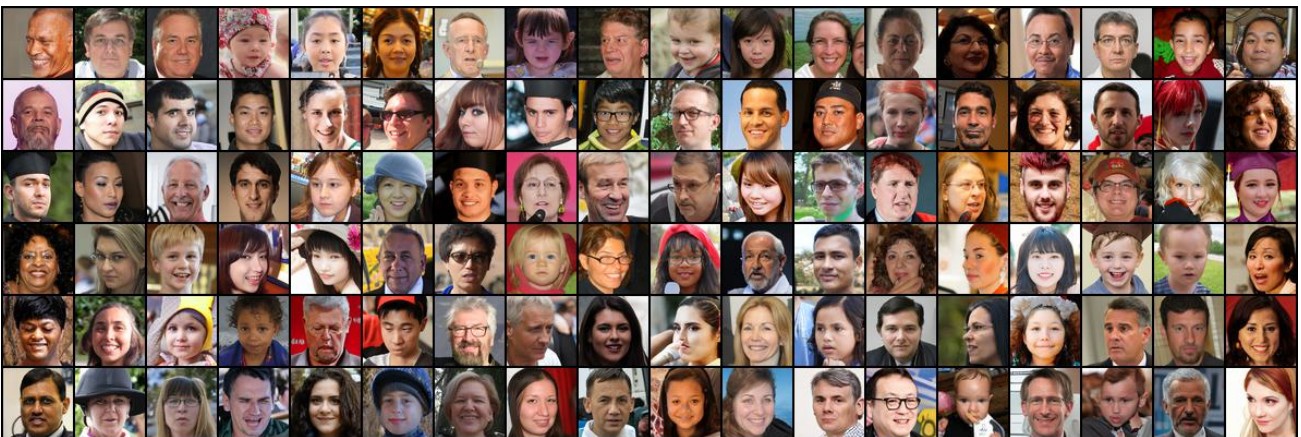

*Figure 17.* FFHQ 64x64, D2O, NFE=1, FID=1.08

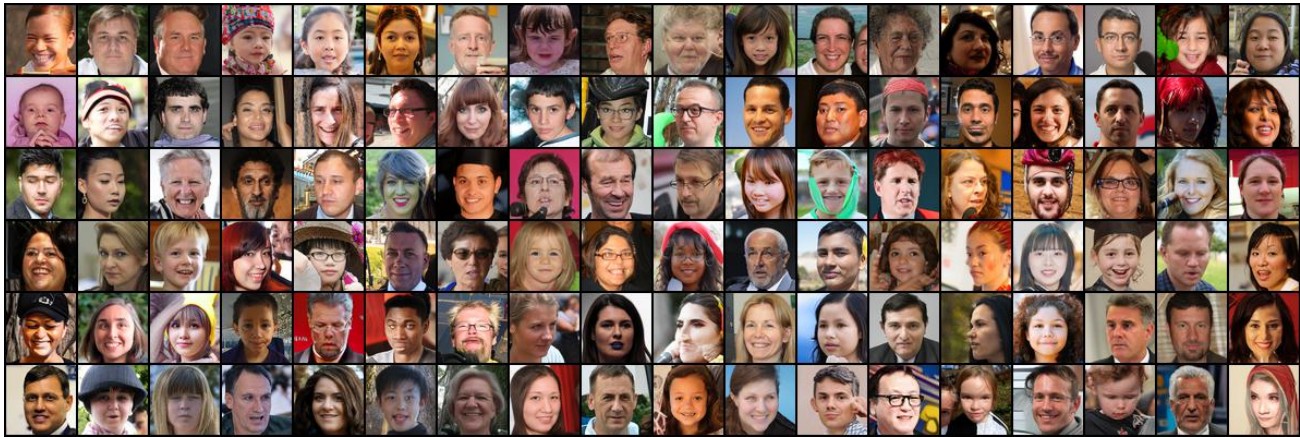

*Figure 18.* FFHQ 64x64, D2O-F, NFE=1, FID=0.85

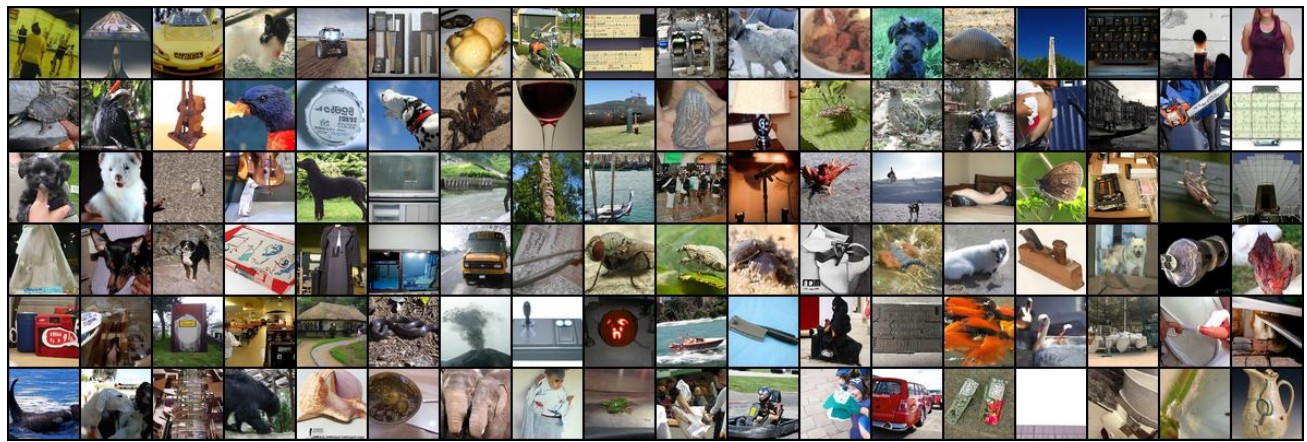

*Figure 19.* ImageNet 64x64 (conditional), EDM (VE), NFE=79, FID=2.36

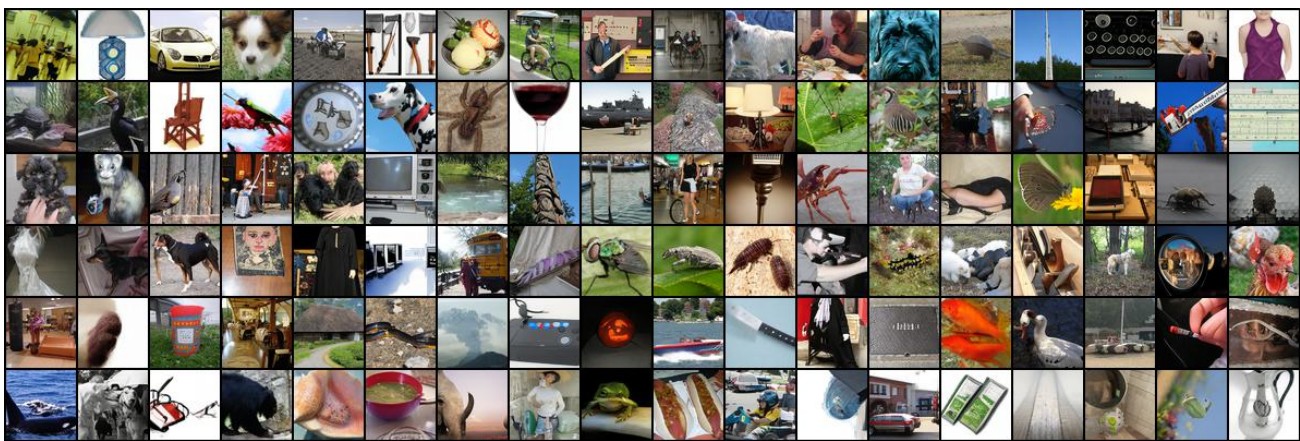

*Figure 20.* ImageNet 64x64 (conditional), D2O, NFE=1, FID=1.42

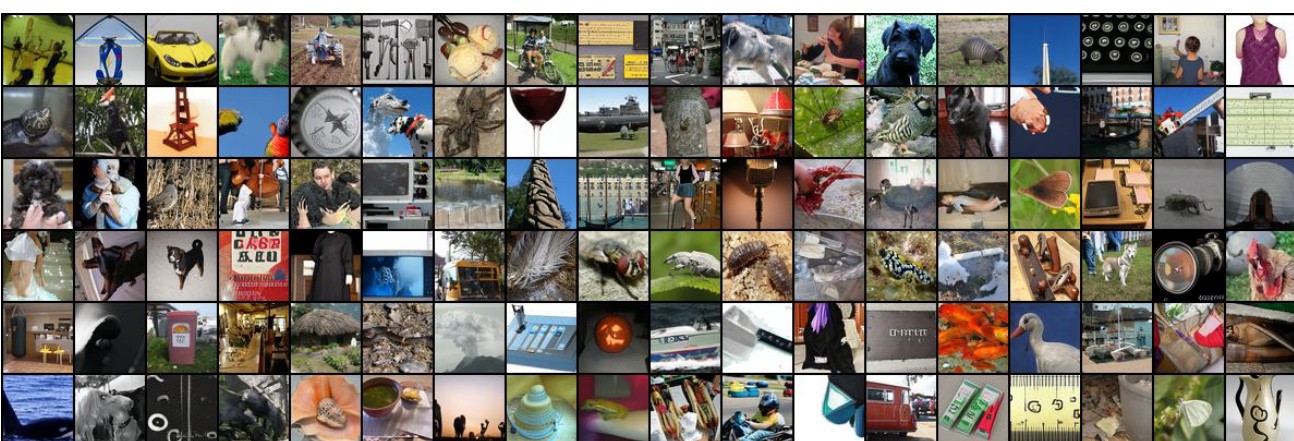

*Figure 21.* ImageNet 64x64 (conditional), D2O-F, NFE=1, FID=1.16

