# OpenReview forum: "Revisiting Diffusion Models: From Generative Pre-training to One-Step Generation"
_ICML.cc/2025/Conference — ICML 2025 poster_

### Official Review · Reviewer_atu8 · 2025-03-11

**Overall Recommendation:** 3

**Summary:**

This work observes that distillation-based training of diffusion models may result in a mismatch of local minima between the student and teacher models. Additionally, it demonstrates that employing a standalone GAN objective, without a distillation objective, is sufficient to transform diffusion models into efficient one-step generators.

## update after rebuttal

I appreciate the authors for their clarifications. After a thorough review of the work and the authors' responses, I have raised my scores. However, I believe that the presentation of the paper could be improved.

**Claims And Evidence:**

The claim that a student model's performance is degraded compared to the teacher when using only a distillation loss is well established in the distillation literature [1,2]. However, the FID plots in Figure 2 do not provide strong empirical support for the assertion regarding the "optimization landscape". Specifically, the statement—"We speculate that this may produce different optimization landscapes and different local minima between the teacher and student model"—lacks direct evidence from the presented results. Moreover, does the phenomena hold for other distillation-based methods?


[1] Sauer et. al., Adversarial Diffusion Distillation

[2] Kim et. al., Consistency Trajectory Model

**Essential References Not Discussed:**

Overall, the authors discuss essential references to some extent. However, I suggest including discussions on the following points:

- The observation that GAN-distillation-based methods may require less training data was previously noted in [1] (see their Section 3.2). Additionally, this reference provides further arguments regarding GAN-based training on top of diffusion models.

- While [1] is a concurrent work, it adopts a similar approach and extends the methodology to video generation. A conceptual discussion of this work would enhance the paper’s contextual positioning.

[1] Ki, et. al., PaGoDA: Progressive Growing of a One-Step Generator from a Low-Resolution Diffusion Teacher

[2] Lin et. al., Diffusion Adversarial Post-Training for One-Step Video Generation

**Experimental Designs Or Analyses:**

The overall validity of the experimental design appears valid and reasonable.

**Methods And Evaluation Criteria:**

The paper employs standard evaluation metrics commonly used in image generation, including FID, IS, and Precision/Recall scores.

**Other Comments Or Suggestions:**

Please refer to the comments above.

**Other Strengths And Weaknesses:**

- Since the proposed method relies solely on GAN loss for training, does this essentially reduce the entire pipeline to standard GAN training, with the pre-trained diffusion model merely serving as a better initialization? If so, could one achieve similar performance by initializing with a pre-trained GAN instead, assuming successful training?

- GAN training is inherently unstable—it requires careful architectural choices and specialized training techniques, as also evidenced in Table 1. Even if the proposed method achieves reasonably good performance, it remains unconvincing compared to distillation-based or consistency training approaches, which offer more stability and theoretical grounding.

- In addition, the FID evaluation of the method is questionable. It is well known that using a GAN loss with a discriminator pretrained on ImageNet significantly biases the FID metric [1]. I suggest evaluating the method using Fréchet distances with DINOv2, following the approach in the EDM2 paper, to provide a more reliable assessment of generative quality.

- Diffusion models and their distilled variants naturally support classifier-free guidance (CFG) by leveraging conditional and unconditional score estimates during sampling. However, a GAN-only distillation framework raises questions about how conditional generation can be effectively incorporated--especially for text-to-image generations.


[1] Kynkäänniemi, et al. The role of imagenet classes in fr'echet inception distance

**Questions For Authors:**

Please refer to the comments above.

**Relation To Broader Scientific Literature:**

This work proposes eliminating the distillation losses commonly used in diffusion distillation methods and relying solely on adversarial loss. The idea is straightforward, and I am concerned about the inherent and classic challenges associated with GAN training.

**Theoretical Claims:**

The paper does not have theoretical results.

---

> ### Author Rebuttal · Authors · 2025-04-01
>
> Thanks for the comments. However, the reviewer may have overlooked some important content of our paper. Below is the point-by-point response.
>
> **Summary:**
>
> Our work is not merely about finding "different local minima" or "distilling diffusion using only a GAN," as the reviewer stated, which only covers the results from Sections 2 and 3. More important contributions are detailed in Sections 4 and 5:
>
> 1. To explain the efficiency of training with a GAN objective, we hypothesize that diffusion training provides a powerful generative capacity and can be viewed as a form of generative pre-training. Under this view, we can bypass iterative trajectory-based sampling/distillation, and a more direct post-training approach can transform diffusion models into one-step generative models with high efficiency (Sections 4.1-4.4).
> 2. We validate this hypothesis by freezing most of the diffusion model’s parameters during training, requiring only minimal fine-tuning (on the order of 0.2M training images). This indicates that our methods leverage the generative pre-training capacity of diffusion models (Sections 4-5).
> 3. Finally, we explore why diffusion models have this general generative capacity with a preliminary frequency analysis, showing they exhibit distinct frequency-specific patterns (Sections 4.5)
>
> **Claims And Evidence:**
>
> The primary concern about the "optimization landscape" appears to reflect a misunderstanding of the results presented in Section 2\. As shown in Figure 2, an increase in the teacher network’s parameters or sampling steps consistently leads to a wider gap in FID scores. This widening gap clearly indicates growing divergence between the learned mappings, suggesting increasingly divergent local optima. This empirical evidence strongly supports our claims of distinct local minima.
>
> Furthermore, Section 2 only serves as an inspiration for us to develop the one-step generation model, which is presented mostly in Sections 4 and 5\. We would like to ask the reviewer to give more consideration to these sections when evaluating the paper.
>
> **Theoretical Claims:**
>
> The reviewer stated that the paper did not have theoretical results. However, we presented theoretical analyses in Section 4.5, which explored the potential mechanism that allows one-step training in our model.
>
> **Supplementary Material:**
>
> We kindly request the reviewer to re-examine this statement: "The authors did not attach supplementary material, although I did browse through the appendix." Detailed supplementary material is indeed included in the manuscript. This is confirmed by other reviewers:
>
> 1. Reviewer BcZC: "Yes, I have reviewed all parts of Supplementary material."
> 2. Reviewer EGMJ: "Yes, checked the implementation details and additional visualizations."
>
> **Other Strengths And Weaknesses:**
>
> 1. "Since the proposed…"
>    This is precisely the main point of our paper, which is mainly explained in Sections 4 and 5, but also stated clearly in the **Introduction (Line 53)**.
>
> 2. "GAN training is inherently unstable…"
>    We discussed this point in **Section 4.2 (Line 249)**: "Notably, with the majority of parameters in both the discriminator and generator frozen, the training process of D2O-F is stable with minimal instances of mode collapse. Therefore, the freezing method circumvents the inherent instability of using GANs."
> 3. "In addition, the FID evaluation of the method is questionable…"
>    We did provide additional CLIP-FID evaluations in **Appendix D** to address this point, which were positively acknowledged by other reviewers.
>
> In summary, we appreciate the reviewer's valuable feedback but would like to kindly request the reviewer to consider re-evaluating the paper and incorporating the analyses and results that were unfortunately overlooked.

---

### Official Review · Reviewer_BcZC · 2025-03-12

**Overall Recommendation:** 3

**Summary:**

This work proposes a method called D2O that fine-tunes a pretrained diffusion model for one-step generation with GAN loss. Pretrained VGG-16 is used as discriminator and the specific GAN loss objective used for the discriminator is Projected GAN (Sauer et al. 2022). In addition, they use other techniques like regularization and normalization to further stabilize training. In addition, they also propose D2O-F that freezes 85% of the parameters in convolutional layers during fine-tuning. This results in a good single-step generator. In addition, the paper also presents some interesting analysis on how different frequencies are processed in diffusion model both within the UNet architecture and during sampling.

**Claims And Evidence:**

1. Writing and overall structure of the paper needs to be improved, as there are instances of contradictory statements. There's also slight overclaiming at certain places which needs to be toned down.
- Consider the following two statements about use of augmentation. There’s clearly some logical disconnect here.
    1. Line 217: We adopt differentiable augmentation (diffAug) without a gradient penalty by default…
    2. Line 190-191 (second column): we find that It leads to poor results in our method… we disable all augmentation in all of our further experiments.
- Contributions are unclear - The initial sections of this paper are devoted towards a method for single step image generation however, towards the end of Section 4, the focus shifts towards understanding how different frequencies are processed during sampling in diffusion models, as well as within the UNet architecture. While the frequency analysis is useful, it feels like a tangential direction to the main goal of this paper, which is single-step image generation.
- Many comparisons are not fair.
    1. “D2O-F produces satisfying images with as few as 0.2 million training steps (FID=4.12). It reaches near SOTA performance by only 5 million steps (FID=1.54). In contrast, training a generative model with similar per- formance typically requires tens or hundreds of millions of training steps (100M for StyleGAN2-ADA, 200M for EDM, on CIFAR-10)” This statement is not fair because D2O-F uses a pre-trained diffusion model (in fact EDM), and the training budget for pretraining needs to be accounted. Similar statements have been made at many other places in the paper. These statements need to be toned down.
2. The experimental set up in Section 2.3 is confusing. The experiment in Section 2.3 is used to argue that teacher and student models converge to different minima in traditional distillation methods, and that the student models fail to match few step teacher predictions. However, only GAN loss is used for fine-tuning both the teacher and student model in these experiments (as specified in Lines 139-140). GAN loss is more suitable for one-step generation and it is unclear why the authors used GAN loss for multistep sampling set up. It is also unclear how the multistep teacher models were trained with only GAN loss. Further, sweeping arguments across all distillation methods have been made based on the results obtained from these fine-tuned models while the set up is quite different from that of traditional distillation methods (See beginning of Section 3.1 for instance). Both the loss landscape and gradient descent dynamics will be quite different for purely GAN-based fine-tuning of pre-trained diffusion models compared to the MSE/Huber loss based distillation methods. Also, if using only GAN loss leads to different minima, as specified in Section 2.3, why would authors still choose to train purely on GAN loss anyway (See Section 3.3)?

**Essential References Not Discussed:**

Related work for consistency models can also be added in the appendix. Many of the recent methods are missing. I have listed these methods above. Check my previous responses for exact references.

**Experimental Designs Or Analyses:**

The experimental designs make sense. I have listed some avenues for improvement of the experimental set up under the previous questions.

**Methods And Evaluation Criteria:**

There are two primary set of results presented in this work.

The first set of results on comparison of quality of images generated by D2O (in terms of FID as well as NFE for efficiency) against the prior methods is detailed. The authors have compared against well-known distillation methods, though I would encourage the authors to add a subset of more recent consistency-based methods like iCT [1], sCT [2], ECT[3], TCM[4] etc. some of which fine-tune from a pretrained diffusion model and might be a more fair comparison. The authors also include results on CLIP-FID in the appendix.

The second  set of results on training sample efficiency feels a bit incomplete. The comparison of training sample efficiency has been made against Consistency Distillation (CD) and SiD which feels a bit incomplete. More precisely, Section 4.1 states: “The amount of data used to train D2O is substantially smaller than that used in most prior distillation methods“ - More evidence is needed for this, especially comparison with the prior distillation based methods e.g. DMD, BOOT, sCM [1], ECT [2], etc, if that data is available.

[1] Song, Yang, and Prafulla Dhariwal. "Improved techniques for training consistency models." arXiv preprint arXiv:2310.14189 (2023).
[2] Lu, Cheng, and Yang Song. "Simplifying, stabilizing and scaling continuous-time consistency models." arXiv preprint arXiv:2410.11081 (2024).
[3] Geng, Zhengyang, et al. "Consistency models made easy." arXiv preprint arXiv:2406.14548 (2024).
[4] Lee, Sangyun, et al. "Truncated Consistency Models." arXiv preprint arXiv:2410.14895 (2024).

**Other Comments Or Suggestions:**

How many fine-tuning steps were used for D2O and D2O-F in Table 4 and 5? Perhaps, this can be indicated in a pair of brackets in the table for easy reference.

**Other Strengths And Weaknesses:**

Strengths:
1. The observation that a (smartly) frozen UNet can be fine-tuned efficienctly with only GAN loss to get good image quality is interesting and beneficial.
2. The frequency analysis of the diffision sampling process as well as the UNet blocks presented both in the main paper and the appendix is interesting (but also seems a bit unrelated to the main problem of one step generation).

Weaknesses: (Please check my previous responses for a detailed feedback on these points.)
1. Quality of writing and overall structure needs improvement.
2. Some baselines are missing and can be added.

**Questions For Authors:**

Is there a typo in Section 2.3 in Lines 139-140 about how the teacher models were fine-tuned with GAN loss for multistep sampling?

**Relation To Broader Scientific Literature:**

There's growing demand for efficient one-step image generators that can generate high quality images quickly. Potentially, the findings here can be extended to videos and other modalities as well.

**Theoretical Claims:**

N/A

---

> ### Author Rebuttal · Authors · 2025-04-01
>
> Thank you for your detailed review and constructive feedback. Here, we provide a point-by-point rebuttal, which we hope helps to clarify the confusion:
>
> **Claims And Evidence:**
>
> 1. "Writing and overall structure…"
>    - "Consider the following two statements about the use of augmentation…"
>       Sorry for the confusion, but there’s no inconsistency here. We first used diffAug as the baseline’s default setting initially. Since the ablation experiments showed that augmentation led to worse performance in our GAN post-training pipeline, we chose not to use augmentation in the rest of the experiments.
>    - "Contributions are unclear…"
>       As you noted, this paper’s main aim is to develop a method for one-step image generation. The frequency analysis in Section 4 is to provide an intuition for understanding why one-step generation with the frozen weights works. We agree that it could belong to Supplementary, but we feel that including it provides a theoretical explanation for our approach, especially for those who do not have time for supplementary materials.
>    - "Many comparisons are not fair…"
>       The point of these comparisons is **not** to argue for a better training efficiency. Instead, they are to demonstrate that our models quickly learn one-step generations by adapting the pre-trained models’ innate capability (since learning from scratch needs far more data than what D2O-F needs), thereby supporting our claim that diffusion training can be viewed as a pre-training for generative capabilities. This seems to be a misunderstanding.
> 2. "The experiment set up in Section 2.3…"
>    Thanks for the constructive criticism. As you correctly noted, our approach differs from traditional trajectory-alignment distillation approaches using L2/Huber losses.
>    However, training a multi-step generator with a GAN loss is entirely feasible. We simply allowed gradients to flow through the entire multi-step sampling process. By employing the same loss function for both teacher and student, the results showing the mismatch between the two (Fig. 2\) are more compelling.
>    Moreover, our goal is not to show that GAN is superior to L2/Huber losses. Rather, we want to demonstrate that the teacher and student networks may converge to different local minima for the same generative task.
>    In this experiment, if we used an L2/Huber loss for trajectory alignment, we would force the student to imitate the teacher, which undermines our goal of letting each model independently learn the target distribution and comparing their solution. Therefore, we used a pure GAN loss as a direct, efficient way to ensure both models optimize the same ultimate objective without additional constraints.
>    Finally, as Fig. 2 supports our hypothesis. Forcing students to replicate the teacher with a L2/Huber trajectory alignment loss could lead to inefficient and performance degradation.  Our central idea proposes to bypass traditional trajectory-based sampling or distillation and to directly tune a pre-trained diffusion model with a simpler objective. Thus, choosing pure GAN is reasonable, which is further solidified by the superior performance of D2O-F. We also tested training with additional L2 (CD) loss (Section 4.4), which did not improve the performance.
>
> **Methods and Evaluation Criteria：**
>
> 1. We admit that the CLIP-FID comparisons in the appendix are limited due to resource constraints. However, we believe that SiD (a near-SOTA method with solid theory and efficiency claims) is a strong baseline to demonstrate our method’s efficiency and prove that our method’s good performance is not due to FID leakage.
> 2. The results in Tables 4 and 5 used 4-7 M training images, depending on the dataset.
> 3. Here, we provide further evidence as requested, with more training images and one-step FID comparison between D2O-F and competing methods on ImageNet 64x64. The name in the quote is the pre-trained diffusion model.
>
>    | Methods | FID | Training Images (Millions) | MParams |
>    | :---- | :---- | :---- | :---- |
>    | BOOT (EDM) | 16.3 | 307  | 280 |
>    | DMD (EDM) | 2.62 | 117  | 280 |
>    | ECM-S (EDM2)  | 5.51 | 12  | 280 |
>    | ECM-S\* (EDM2)  | 4.05 | 102 | 280 |
>    | ECM-XL (EDM2) | 3.35 | 12  | 1119 |
>    | ECM-XL\* (EDM2) | 2.49 | 102  | 1119 |
>    | sCD-S (EDM2+TrigFlow) | 2.97 | 819 | 280 |
>    | sCD-XL (EDM2+TrigFlow) | 2.44  | 819 | 1119 |
>    | **D2O-F** (EDM) | **1.16** | **5** | 280 |
> Again, we’d like to thank the reviewer for the time and effort. With the additional results and the clarification, we sincerely hope the reviewer would re-evaluate the paper accordingly.

---

> > ### Comment · Reviewer_BcZC · 2025-04-04
> >
> > I thank the authors for their response. I'm satisfied with the response and therefore adjusting my score. I would however still request the authors to restructure the paper so that the flow is more logical. More specifically, consider introducing some insights from the frequency analysis early on so that reader expects this later in the paper. Further, highlight the main goal of this frequency analysis which is to provide an intuition for understanding why one-step generation with the frozen weights works. Currently, this is not sufficiently highlighted. Further, consider toning down certain statements which still feel like overclaiming. The method nonetheless seems to have good results with good training efficiency. Therefore I'm increasing my score.

---

> > > ### Author Response · Authors · 2025-04-07
> > >
> > > Thanks for your recognition and comments. We will make the appropriate revisions in the camera-ready version, including clarifying the purpose and significance of the frequency analysis and revising the results to reflect your points.

---

### Official Review · Reviewer_EGMJ · 2025-03-13

**Overall Recommendation:** 3

**Summary:**

The paper proposes a novel approach, D2O (Diffusion to One-Step), which uses a GAN objective to convert diffusion models (DMs) into efficient one-step generators. It identifies a key issue in previous distillation methods: the teacher and student models' distinct local minima, which enables effective knowledge transfer. The paper argues that a standalone GAN objective can bypass this issue, enabling the diffusion model to be fine-tuned for one-step generation without relying on distillation losses. The authors introduce D2O-F, where most parameters are frozen during fine-tuning, further improving the method's efficiency. Through experiments on datasets like CIFAR-10 and ImageNet, they show that D2O and D2O-F achieve competitive performance with fewer training images compared to traditional methods.

**Claims And Evidence:**

The claims are generally supported by clear and convincing evidence. The authors show that D2O and D2O-F achieve competitive results on several datasets (CIFAR-10, ImageNet, etc.) with much fewer training images (0.2M to 5M). The use of a GAN objective during fine-tuning is demonstrated to overcome the local minima problem in diffusion distillation while improving efficiency. The freezing experiment further validates the claim that diffusion models provide sufficient generative capabilities through pre-training, requiring minimal adjustments during fine-tuning. The comparison with previous distillation methods also supports the proposed approach's effectiveness.

**Essential References Not Discussed:**

N/A

**Experimental Designs Or Analyses:**

The experimental designs are sound, particularly the ablation study of the D2O-F model, which freezes most parameters and shows that this strategy enhances performance. The authors also test D2O and D2O-F on multiple datasets and tasks, confirming their results across different settings. As the authors acknowledged, however, the freezing method's effect on more complex architectures (e.g., DiT) and higher-resolution datasets has not been tested, which might limit the generalizability of the findings.

**Methods And Evaluation Criteria:**

The proposed method and evaluation criteria (FID, Inception Score, Precision, and Recall) are standards for generative modeling tasks. The use of multiple datasets (CIFAR-10, AFHQv2, FFHQ, ImageNet) provides a broad assessment of the model's performance.

**Other Comments Or Suggestions:**

N/A

**Other Strengths And Weaknesses:**

By using the freezing strategy (freezing most of the pre-trained parameters during fine-tuning), the paper demonstrates that you can achieve competitive performance with much fewer training steps (as few as 0.2M training steps) compared to traditional methods, which require tens or even hundreds of millions of training steps.

The NFE results are mentioned in the tables, but there's a lack of detailed analysis on them or discussion regarding the inference gains.

**Questions For Authors:**

N/A

Update after rebuttal: After reviewing the rebuttal discussions, I would keep the original rating.

**Relation To Broader Scientific Literature:**

The paper positions itself as an advancement over existing methods by addressing the limitations of distillation, providing an efficient method for one-step generation. The relationship to GAN-based distillation methods (e.g., Sauer et al., 2023) is also highlighted. The proposed approach offers a promising alternative to multi-step distillation in generative modeling, which is a current challenge in the field.

**Theoretical Claims:**

The paper makes a claim that the local minima of teacher and student models differ significantly, challenging the distillation process. And it also claims that the GAN objective leads to better convergence compared to traditional distillation. No formal proof of these theoretical claims are provided, but the empirical results look reasonable to me.

---

> ### Author Rebuttal · Authors · 2025-04-01
>
> Thank you for your detailed and positive feedback. We acknowledge that our current work is primarily empirical. Nevertheless, we believe that our work provides important insights for the community, particularly regarding how to take advantage of the capabilities within pre-trained diffusion models. We validate this idea with experiments on the one-step generation task. We plan to strengthen the theoretical foundations and extend the method to more complex datasets.

---

### Decision · Program_Chairs · 2025-05-01

**Decision:**

Accept (poster)

**Comment:**

This paper presents a novel and empirically validated approach—D2O and D2O-F—to convert diffusion models into efficient one-step generators via GAN-based fine-tuning, reframing diffusion pretraining as a powerful generative foundation. The authors identify and empirically support a key limitation in prior distillation-based methods: the mismatch of local minima between teacher and student models due to architectural and procedural differences. By eliminating the need for trajectory-aligned distillation losses and instead relying solely on adversarial objectives, the proposed method significantly reduces inference complexity while maintaining strong generative performance.

strengths:
* demonstrates state-of-the-art or near-SOTA FID scores on multiple datasets with drastically fewer training images.
* introduces a compelling perspective that diffusion training acts as generative pretraining.
* the freezing technique (D2O-F) shows that only minimal parameter tuning is needed to achieve high-quality results, enhancing stability and efficiency.
* reviewers appreciate the breadth of experiments, ablations, and the initial frequency analysis explaining the efficacy of one-step generation.

weaknesses:
* reviewers highlighted presentation issues (e.g., writing clarity, structural flow, some overclaiming), but the authors addressed these in the rebuttal and committed to improving them.
* clarifications were requested around training setups and comparison fairness, especially regarding pretraining budgets, which the authors provided in detail.
* reviewer atu8 raised concerns about generality, GAN instability, and evaluation biases. However, many points appear to stem from partial engagement with the full content, and the authors responded with clarifying evidence and results that were acknowledged by other reviewers.

While not without imperfections, the paper introduces a novel and promising method that offers both practical impact and conceptual insight into diffusion-based generative modeling. Therefore it would benefit the community to further discuss the work.